# Pareto Deep Long-Tailed Recognition: A Conflict-Averse Solution

**Zhipeng Zhou**[1], **Liu Liu**[2,†], **Peilin Zhao**[2], **Wei Gong**[1,†]
[1]University of Science and Technology of China, [2]Tencent AI Lab
zzp1994@mail.ustc.edu.cn, {leonliuliu, masonzhao}@tencent.com
weigong@ustc.edu.cn

## Abstract

Deep long-tailed recognition (DLTR) has attracted much attention due to its close touch with realistic scenarios. Recent advances have focused on re-balancing across various aspects, e.g., sampling strategy, loss re-weighting, logit adjustment, and input/parameter perturbation, etc. However, few studies have considered dynamic re-balancing to address intrinsic optimization conflicts, which are identified as prevalent and critical issues in this study. In this paper, we empirically establish the severity of the optimization conflict issue in the DLTR scenario, which leads to a degradation of representation learning. This observation serves as the motivation for pursuing Pareto optimal solutions. Unfortunately, a straightforward integration of multi-objective optimization (MOO) with DLTR methods is infeasible due to the disparity between multi-task learning (MTL) and DLTR. Therefore, we propose effective alternatives by decoupling MOO-based MTL from a temporal perspective rather than a structural one. Furthermore, we enhance the integration of MOO and DLTR by investigating the generalization and convergence problems. Specifically, we propose optimizing the variability collapse loss, guided by the derived MOO-based DLTR generalization bound, to improve generalization. Additionally, we anticipate worst-case optimization to ensure convergence. Building upon the proposed MOO framework, we introduce a novel method called **P**areto deep **LO**ng-**T**ailed recognition (`PLOT`). Extensive evaluations demonstrate that our method not only generally improves mainstream pipelines, but also achieves an augmented version to realize state-of-the-art performance across multiple benchmarks. Code is available at `https://github.com/zzpustc/PLOT`.

## 1 Introduction

Nowadays success of machine learning (ML) techniques are largely attributed to the growing scale of the training dataset, as well as the assumption of it being independent and identically distributed (i.i.d) with the test distribution. However, such an assumption can hardly hold in many realistic scenarios where training sets show an imbalanced or even long-tailed distribution, raising a critical challenge to the traditional ML community (Zhang et al., 2021b). To address this issue, recent researches devoted on deep long-tailed recognition (DLTR) has gained increasing interests, which strives to mitigate the bias toward certain categories and generalize well on a balanced test dataset.

Plenty of approaches have been proposed to realize re-balancing from various aspects in DLTR (Zhang et al., 2021b): sampling strategy (Zang et al., 2021; Cai et al., 2021), loss function (Wang et al., 2013; Ren et al., 2020; Tan et al., 2020), logit adjustment (Cao et al., 2019; Li et al., 2022), data augmentation (Kim et al., 2020; Wang et al., 2021), input/parameter perturbation (Rangwani et al., 2022; Zhou et al., 2023), decoupling learning regime (Kang et al., 2019), and diverse experts (Wang et al., 2020b; Guo & Wang, 2021), etc. Usually, these works design fixed re-balancing strategies according to the prior of the class frequency to ensure all categories are generally equally optimized.

Several very recent researches (Ma et al., 2023; Sinha & Ohashi, 2023; Tan et al., 2023) empirically indicate that a dynamic re-balancing strategy is required, and achieve it by designing a quantitative

---

† Corresponding authors. Work done when Z. Zhou works as an intern in Tencent AI Lab.

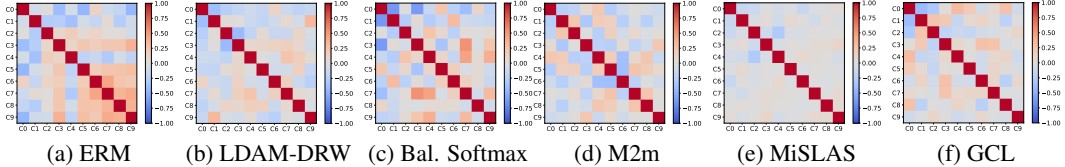

(a) ERM     (b) LDAM-DRW  (c) Bal. Softmax  (d) M2m     (e) MiSLAS     (f) GCL

Figure 1: Gradient conflicts among categories. 'Bal. Softmax' is short for Balanced Softmax. The horizontal and vertical coordinates are for each category and the heat map represents the gradient similarity.

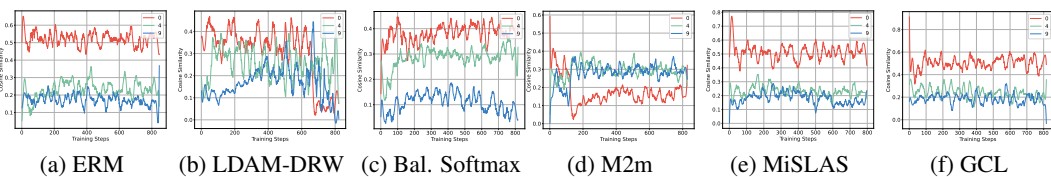

(a) ERM     (b) LDAM-DRW  (c) Bal. Softmax  (d) M2m     (e) MiSLAS     (f) GCL

Figure 2: Gradient similarities during optimization. '0', '4', and '9' denote the corresponding categories in CIFAR10-LT, belonging to the 'head', 'medium', 'tail' classes. Please refer to the gradient norm examination in **Section 4.5 of the Appendix**.

measurement of semantic scale imbalance and a meta module to learn from logit, etc. All these works take the instant rather than prior imbalance into consideration, enabling them to reach a competitive performance across various imbalanced scenarios. Nevertheless, a question is naturally raised:

*Is instant imbalance enough for designing a dynamic re-balancing strategy?*

We then delve into the optimization of representative DLTR models and present related observations from the perspective of multi-objective optimization (MOO) in Fig. 1 and Fig. 2. As depicted, intrinsic optimization conflicts among categories are prevalent and might be aggravated due to the dominated trajectories of certain categories, which would lead to sub-optimal solutions for the remaining ones. Such an issue is rarely discussed for the above question and cannot be addressed by current dynamic strategies due to their lack of design nature (Refer to Section 3 for more details).

To fill this gap, we approach DLTR from a new angle, i.e., mitigate optimization conflicts via dynamic re-balancing, which is usually neglected in past works. We first identify the existing intrinsic gradient conflicts among categories in the optimization of prevailing DLTR methods and show their connection with the adopted fix re-balancing strategies. To prevent the representation from being overwhelmed by dominated categories' properties, we introduce MOO in MTL to mine the shared features among categories. Unfortunately, a naïve combination is not applicable due to the structure difference between MTL and DLTR as illustrated in Fig. 6. Specifically, MOO-based MTL usually assumes that the model architecture is consist of a backbone network and several separate task-specific branches on top of the backbone network, and strives to learn task-shared features with backbone network via MOO algorithms. While DLTR targets only one task and owns only one branch. Hence a critical challenge appears:

*How to engage MOO into DLTR?*

As depicted in Fig. 6, we tackle this challenge with two key enablers: (1) Regarding a multi-classification task as multiple binary classification tasks, (2) transform the shared feature extraction and task-specific optimization from structural to temporal. Besides, by investigating several popular MOO approaches and choosing a stable one, we provide instructions on the integration of DLTR and MOO, propose variability collapse loss, and anticipate worst-case optimization to ensure generalization and convergence. It should be noted that our goal is to provide a distinct angle of re-balancing rather than design a new instant imbalance metric or MOO method, thus comparing our approach with these counterparts is beyond the scope of this paper.

**Contributions**: Our contributions can mainly be summarized as four-fold:

- Through the lens of MOO, we empirically identify the phenomena of optimization conflicts among categories and establish its severity for representation learning in DLTR.

- To mitigate the above issues, we endow prevailing re-balancing models with Pareto property via innovatively transforming the MOO-based MTL from structural to temporal, enabling the application of MOO algorithm in DLTR without model architecture modifications.

- Moreover, two theoretical motivated operations, i.e., variability collapse loss, and anticipating worst-case optimization are proposed to further ensure the generalization and convergence of MOO-based DLTR.

- Extensive evaluations have demonstrated that our method, PLOT, can significantly enhance the performance of mainstream DLTR methods, and achieve state-of-the-art results across multiple benchmarks compared to its advanced counterparts.

## 2 PRELIMINARIES

**Problem Setup:** Taking a $K$-way classification task for example, assume we are given a long-tailed training set $\mathcal{S} = (\boldsymbol{x_i}, \boldsymbol{y_i} | i = 1, \ldots, n)$ for the DLTR problem. And the corresponding per-class sample numbers are $\{n_1, n_2, ..., n_K\}, n = \sum_i^K n_i$. Without loss of generality, we assume $n_i < n_j$ if $i < j$, and usually $n_K \gg n_1$. Following the general DLTR setting, all models are finally evaluated on a balanced test dataset.

**Pareto Concept:** Our framework hinges on MOO-based MTL, which strives to achieve the Pareto optimum under the MTL situation. Formally, assume that there are $N$ tasks at hand, and their differentiable loss functions are $\mathcal{L}_i(\boldsymbol{\theta}), i \in [N]$. The weighted loss is $\mathcal{L}_{\boldsymbol{\omega}} = \sum_{i=1}^N \omega_i \mathcal{L}_i(\boldsymbol{\theta}), \boldsymbol{\omega} \in \mathcal{W}$, where $\boldsymbol{\theta}$ is the parameter of the model and $\mathcal{W}$ is the probability simplex on $[N]$. A point $\boldsymbol{\theta'}$ is said to Pareto dominate $\boldsymbol{\theta}$, only if $\forall i, \mathcal{L}_i(\boldsymbol{\theta'}) \leq \mathcal{L}_i(\boldsymbol{\theta})$. And therefore the Pareto optimal is the situation that no $\boldsymbol{\theta'}$ can be found that holds $\forall i, \mathcal{L}_i(\boldsymbol{\theta'}) \leq \mathcal{L}_i(\boldsymbol{\theta})$ for the point $\boldsymbol{\theta}$. All points that satisfy the above conditions are called Pareto sets, and their solutions are so-called Pareto fronts. Another concept called Pareto stationary, which requires $\min_{\boldsymbol{\omega} \in \mathcal{W}} \|\boldsymbol{g_\omega}\| = 0$, where $\boldsymbol{g_\omega}$ is the weighted gradient. In this paper, since we regard the $K$-way classification task as $K$ binary tasks, $N$ is assigned as $K$.

**Definition 2.1** (**Gradient Similarity**). *Denote $\phi_{ij}$ as the angle between two task gradients $\boldsymbol{g_i}$ and $\boldsymbol{g_j}$, then we define the gradient similarity as $\cos \varphi_{ij}$ and the gradients as conflicting when $\cos \phi_{ij} < 0$.*

**Definition 2.2** (**Dominated Conflicting**). *For task gradients $\boldsymbol{g_i}$ and $\boldsymbol{g_j}$, assume their average gradient as $\boldsymbol{g_0}$, and $\|\boldsymbol{g_i}\| < \|\boldsymbol{g_j}\|$. Then we define the gradients as dominated conflicting when $\cos \phi_{0i} < 0$.*

## 3 MOTIVATION AND EMPIRICAL OBSERVATIONS

**Intrinsic Property in Definition and Difference with MTL**: As outlined in Section 2, a DLTR model is trained on an imbalanced dataset but is expected to generalize well to all categories, which aligns with the motivation of Pareto optimality, i.e., improving all individual tasks (categories). However, unlike MTL, which employs a distinct structure where the backbone and corresponding branches are responsible for shared feature extraction and task-specific optimization, respectively, the structures in DLTR are attributed to all categories. This difference impedes DLTR models from achieving Pareto properties. Therefore, in Section 4, we introduce the MOO-based DLTR pipeline. **Optimization Conflicts under Imbalanced Scenarios**: As depicted in Fig. 3, it is evident that

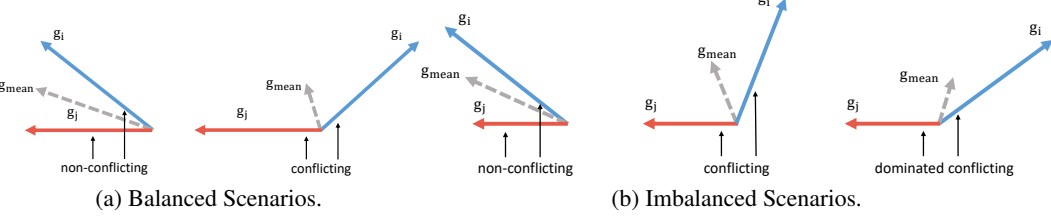

(a) Balanced Scenarios.    (b) Imbalanced Scenarios.

Figure 3: Illustration of gradient conflict scenarios.

each task exhibits improvement when optimized using the average gradient, i.e., $g_{mean}$, in balanced

Table 1: Benefits of MOO methods for mainstream DLTR models on CIFAR10-LT. We re-implement all models via their publicly released code, and all results are reported over 3 random seeds experiments. ▲/▼ indicates outperforms/underperforms their vanilla versions, while the early stop version is colored and the naïve integration version is underlined.

| Imb. | cRT+Mixup | | | | LDAM-DRW | | | |
|---|---|---|---|---|---|---|---|---|
| | Vanilla | w/ EPO | w/ MGDA | w/ CAGrad | Vanilla | w/ EPO | w/ MGDA | w/ CAGrad |
| 200 | 73.06 | 33.45 76.24▲ | 68.05 75.98▲ | 75.15 76.02▲ | 71.38 | 56.04 73.64▲ | 67.18 74.08▲ | 55.80 73.28▲ |
| 100 | 79.15 | 34.27 79.69▲ | 73.71 79.26▲ | 79.58 80.16▲ | 77.71 | 66.49 77.25▲ | 73.70 77.79▲ | 66.49 76.86▼ |
| 50 | 84.21 | 36.53 83.79▼ | 79.27 84.15▲ | 83.52 84.49▲ | 81.78 | 72.60 81.62▼ | 78.24 81.58▼ | 69.26 81.85▲ |

| Imb. | Balanced Softmax | | | | M2m | | | |
|---|---|---|---|---|---|---|---|---|
| | Vanilla | w/ EPO | w/ MGDA | w/ CAGrad | Vanilla | w/ EPO | w/ MGDA | w/ CAGrad |
| 200 | 81.33 | 45.37 81.40▲ | 74.13 80.90▼ | 79.20 80.93▼ | 73.43 | 51.90 73.07▼ | 57.14 72.63▼ | 70.95 73.84▲ |
| 100 | 84.90 | 44.33 85.30▲ | 79.06 85.10▲ | 83.77 85.40▲ | 77.55 | 57.89 76.57▼ | 52.37 76.48▼ | 76.24 77.95▲ |
| 50 | 89.17 | 41.43 88.97▼ | 79.43 88.90▼ | 88.00 89.27▲ | 80.94 | 42.07 81.19▲ | 46.38 80.66▼ | 78.19 81.11▲ |

| Imb. | MiSLAS | | | | GCL | | | |
|---|---|---|---|---|---|---|---|---|
| | Vanilla | w/ EPO | w/ MGDA | w/ CAGrad | Vanilla | w/ EPO | w/ MGDA | w/ CAGrad |
| 200 | 76.59 | 36.62 76.97▲ | 63.40 76.12▼ | 76.30 77.43▲ | 79.25 | 62.08 79.73▲ | 75.43 80.03▲ | 78.73 80.08▲ |
| 100 | 81.33 | 39.92 81.22▼ | 68.09 82.00▲ | 82.10 82.47▲ | 82.85 | 74.78 82.75▼ | 79.01 82.81▼ | 82.48 83.48▲ |
| 50 | 85.23 | 44.78 84.60▼ | 70.20 84.84▼ | 85.20 85.33▲ | 86.00 | 78.42 84.55▼ | 81.89 85.58▼ | 85.31 85.90▼ |

scenarios where conflicts arise. However, in imbalanced scenarios, the utilization of $g_{mean}$ tends to favor the dominant tasks. This preference becomes particularly pronounced in extreme cases, where even when $g_{mean}$ and $g_j$ are in conflict (referred to as Dominated Conflicting in Definition 2.2), the optimization of task $i$ leads to an enhancement in its performance at the expense of task $j$. In order to investigate the presence of optimization conflict issues in DLTR, we meticulously analyze several re-balancing regimes: (1) Cost-sensitive loss approaches, such as LDAM-DRW (Cao et al., 2019) and BalancedSoftmax (Ren et al., 2020); (2) Augmentation techniques, such as M2m (Kim et al., 2020); and (3) Decoupling methods, including cRT + Mixup (Kang et al., 2019), MiSLAS (Zhong et al., 2021), and GCL (Li et al., 2022). By computing the cosine similarities among gradients (referred to as Gradient Similarity in Definition 2.1) associated with different categories, we illustrate the conflict status of these methods in Fig. 1 [1]. As depicted, all the selected methods exhibit varying degrees of gradient conflicts, which persist in the early stages of training (refer to **Section 4.2 in the Appendix**). Furthermore, we examine the optimization preference of DLTR models in Fig. 2, revealing that certain categories dominate the overall optimization process. Additionally, we provide statistical analysis on the frequency of dominated conflicting instances in Fig. 5, establishing a roughly positive correlation between the frequency of dominated conflicting and the imbalance ratio.

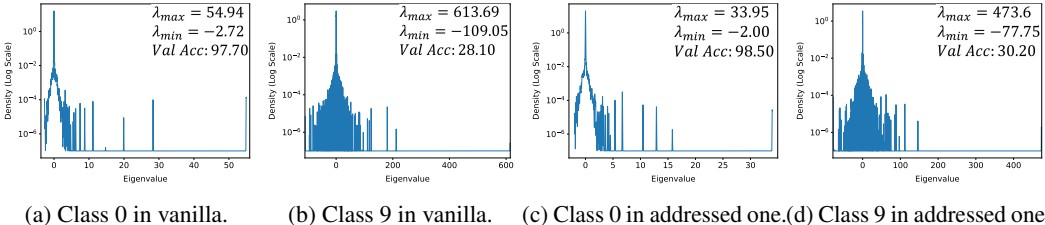

(a) Class 0 in vanilla.  (b) Class 9 in vanilla.  (c) Class 0 in addressed one.(d) Class 9 in addressed one.

Figure 4: Hessian spectrum analysis of before and after addressing optimization conflict issue.

**The Benefit of Addressing Optimization Conflicts**: Here we provide a preview of the advantages achieved by addressing optimization conflicts through the utilization of our temporal design, as outlined in Section 4.1. We present the benefits from two perspectives: (1) representation analysis and (2) performance improvements. To gain insights into the impact of addressing the optimization conflict issue on representation learning, we conducted a Hessian spectrum analy-

---

[1]Mainstream DLTR methods usually employ the SGD optimizer for implementation. Therefore, our analysis does not encompass the results obtained by utilizing alternative optimizers such as Adam.

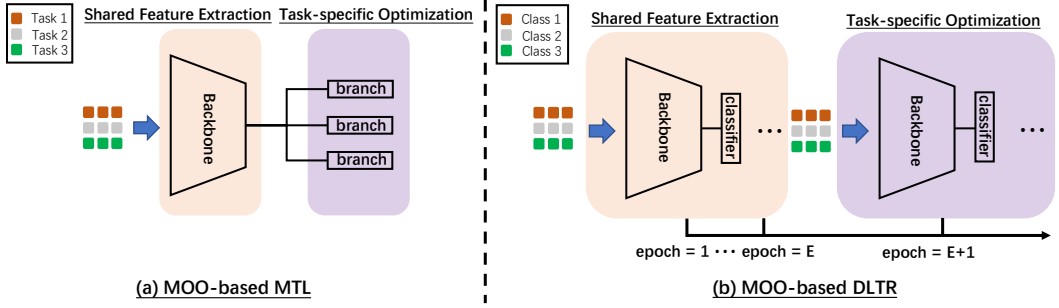

Figure 6: Comparison of MOO-based MTL and MOO-based DLTR.

sis (Rangwani et al., 2022) and visualized the results in Fig. 4. Our analysis reveals that addressing optimization conflicts results in flatter minima for each class, thereby mitigating the risk of being trapped at saddle points and facilitating the acquisition of more generalized representations. Furthermore, we demonstrate the performance benefits achieved by employing various MOO approaches in Table 1. The results effectively showcase the potential of integrating MOO with DLTR.

**Urgency of Conflict-Averse Strategy**: Currently, our MOO-based DLTR framework can primarily be classified as a specialized dynamic re-balancing strategy, formally formulated as $\mathcal{L}(\boldsymbol{x}, \boldsymbol{y}) = \frac{\sum_{k=1}^{K} \omega_k \cdot B_j \cdot \bar{l}(\boldsymbol{x}_*^k, \boldsymbol{y}_*^k)}{B}$. Here $B$ is the batch size, $B_*$ and $\omega_*$ represent the frequency and dynamic re-weighting factor of class $*$, respectively, and $\bar{l}(\boldsymbol{x}_*^k, \boldsymbol{y}_*^k)$ is the average loss of class $*$. While there are existing studies that explore the concept of dynamic re-balancing (Tan et al., 2023; Sinha & Ohashi, 2023; Ma et al., 2023), none of them address the issue of optimization conflicts from a comprehensive perspective. Consequently, the intrinsic optimization conflicts among categories cannot be effectively mitigated (See **Section 4.4 in the Appendix**). Fortunately, our work bridges this gap and offers a solution to this problem.

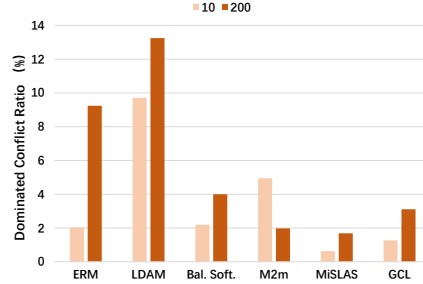

Figure 5: Statistical of dominated conflicts.

## 4 PARETO DEEP LONG-TAILED RECOGNITION

Building on the aforementioned analysis, we present a detailed design for integrating MOO into DLTR, which encompasses its adaptation from MTL to DLTR and an augmented version to ensure generalization and convergence. For complete proofs of the proposed theorems, please refer to **Section 1 of the Appendix**.

### 4.1 MOO: FROM MTL TO DLTR

As stated previously, a straightforward integration of MOO is not feasible for DLTR scenarios due to differences in task properties and architectures. Upon revisiting the function of each component in MOO-based MTL, they can be categorized into two aspects: (1) Shared feature extraction (SFE) and (2) Task-specific optimization (TSO). SFE aims to extract shared representations among distinct tasks, while TSO is responsible for the corresponding task performance with the independent branch. In this study, we approach the multi-classification task by treating it as multiple binary classification tasks. Each binary classification task is considered as a single objective in MOO, and we re-design SFE and TSO from a temporal perspective during the training stage, rather than a structural one. Specifically, we implement SFE in the first $E$ epochs by applying MOO algorithms in DLTR models, but release it in the subsequent stages, as illustrated in Fig. 6. This approach can also be interpreted as an early stopping operation, inspired by previous research (Cadena et al., 2018; Hu et al., 2021) suggesting that the early layers of neural networks undergo fewer changes in the later stages of training. To validate the effectiveness of this design, we employ three representative MOO algorithms, i.e., MGDA (Désidéri, 2012), EPO (Mahapatra & Rajan, 2020), and CAGrad (Liu et al., 2021a), to

equip them on the aforementioned six DLTR models, and the results are presented in Table. 1. The results demonstrate that our design enables MOO to enhance DLTR models in most cases, which also highlights the potential benefits of addressing the optimization conflict problem. Moreover, the performance would significantly deteriorate without an early stop (naïve integration version), indicating class-specific feature degradation. It is also noteworthy that CAGrad exhibits a relatively stable performance across various baselines, thus we select it as the fundamental framework for further enhancement[2].

**Remark** (No Modifications on Model Architecture). *Once again, it is important to emphasize that our approach does not involve any modifications to the model architecture. Rather, our method represents an effective learning paradigm for the application of MOO in DLTR.*

### 4.2 CHALLENGES OF CAGRAD UNDER DLTR

**Generalization Problem**: Prior to delving into the additional technical designs of `PLOT`, it is necessary to establish the learning guarantees for the MOO-based DLTR framework. To this end, we introduce $\omega_k$ into the definition of Rademacher complexity (Cortes et al., 2020), which can be reformulated as follows:

$$\mathfrak{R}_S(\mathcal{G}, \omega) = \mathbb{E}_{\sigma}\left[\sup_{h \in \mathcal{H}} \sum_{k=1}^{K} \omega_k \frac{1}{m_k} \sum_{i=1}^{m_k} \sigma_i \cdot l(h(x_i^k), y_i^k)\right] \tag{1}$$

where $\mathcal{G}$ is associated to the hypothesis set as $\mathcal{H} : \{\mathcal{G} : (x, y) \mapsto l(h(x), y) : \forall h \in \mathcal{H}\}$; $\sigma_i$ is the independent uniformly distributed random variables taking values in $\{-1, +1\}$. From this definition, we have the following theorem:

**Theorem 4.1.** *(MOO-based DLTR Generalization Bound) If the loss function $l_k$ belonging to $k_{th}$ category is $M_k$-Lipschitz, and $\forall(x, y), (x', y') \in \mathcal{X} \times \mathcal{Y}, \forall h \in \mathcal{H}: \|[h(x), y] - [h(x'), y']\| \leq \mathcal{D}_{\mathcal{H}}$, assume $M_k \mathcal{D}_{\mathcal{H}}$ is bounded by $M$, then for any $\epsilon > 0$ and $\delta > 0$, with probability at least $1 - \delta$, the following inequality holds for $\forall h \in \mathcal{H}$ and $\forall \omega \in \mathcal{W}$:*

$$\mathcal{L}_{\omega}(h) \leq \hat{\mathcal{L}}_{\omega}(h) + 2\mathfrak{R}_S(\mathcal{G}, \omega) + \sum_{k=1}^{K} \omega_k M \sqrt{\frac{\log \frac{1}{\delta}}{2}}$$

The above derived generalization bound indicates that we should minimize $\hat{\mathcal{L}}_{\omega}(h)$ as well as constrain the intra-class loss variability $M$. With this theoretical insight, we design the following **variability collapse loss**:

$$\mathcal{L}_{vc} = \frac{1}{K} \sum_{k=1}^{K} \text{Std}(\widetilde{l}(x_*^k, y_*^k)) \tag{2}$$

where $\text{Std}(\cdot)$ is the standard deviation function and $\widetilde{l}(x_*^k, y_*^k)$ is the loss set of the $k_{th}$ category in a mini-batch. It is worth noting that our proposed design shares a similar concept with a recent study (Liu et al., 2023), which aims to induce *Neural Collapse* in the DLTR setting. However, our approach is distinct in that we propose it from the perspective of MOO with theoretical analysis.

**Convergence Problem**: On the other hand, although CA-Grad exhibits stability, it may not always yield improvements, as evidenced by Table 1. Consequently, we delve deeper into CAGrad and demonstrate its limitations in the DLTR scenario. Based on the convergence analysis of CAGrad, we present the following theorem:

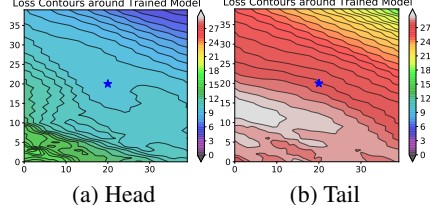

(a) Head          (b) Tail

Figure 7: Loss landscape of LDAM.

**Theorem 4.2.** *(Convergence of CAGrad in DLTR) With a fix step size $\alpha$ and the assumption of H-Lipschitz on gradients, i.e., $\|\nabla \mathcal{L}_i(\theta) - \nabla \mathcal{L}_i(\theta')\| \leq H \|\theta - \theta'\|$ for $i = 1, 2, ..., K$. Denote $d^*(\theta_t)$ as the optimization direction of CAGrad at step t, then we have:*

$$\mathcal{L}(\theta_{t+1}) - \mathcal{L}(\theta_t) \leq -\frac{\alpha}{2}(1 - c^2) \|g_0(\theta_t)\|^2 + \frac{\alpha}{2}(H\alpha - 1) \|d^*(\theta_t)\|^2,$$

---

[2]Nonetheless, the objective of this paper is to explore the potential of the MOO framework in addressing the DLTR problem and to propose an effective algorithm for enhancing mainstream DLTR methods. Therefore, we defer the development and integration of more advanced MOO algorithms to future research.

where $c \in (0, 1)$ and $\boldsymbol{g_0}(\boldsymbol{\theta_t})$ is the corresponding average gradient of $\boldsymbol{\theta_t}$. The convergence of CAGrad is inherently influenced by the value of $H$, as observed in our experiments. This observation is further supported by a related study (Fernando et al., 2022), which introduces random noise to the optimization trajectory of CAGrad and demonstrates convergence failures. Additionally, it is widely acknowledged that achieving a small value of $H$ for tail classes in DLTR models is often unattainable (Rangwani et al., 2022; Zhou et al., 2023). To visually illustrate this point, we provide a depiction of the LDAM-DRW loss landscape in Fig. 7 (for more visual illustrations, please refer to **Section 7.2 in the Appendix**). In this particular case, the loss landscape of the tail class exhibits a sharp minimum, indicating a large value of $H$ and consequently posing challenges for achieving convergence when integrating CAGrad with DLTR models. To solve this problem, we constrain $H$ by **anticipating worst-case optimization** (Foret et al., 2020), i.e., Sharpness aware minimization (SAM):

$$\min_{\boldsymbol{\theta}} \max_{\boldsymbol{\epsilon}(\boldsymbol{\theta})} \mathcal{L}(\boldsymbol{\theta} + \boldsymbol{\epsilon}(\boldsymbol{\theta})), \text{ where } \|\boldsymbol{\epsilon}(\boldsymbol{\theta})\|_2 \leq \rho, \tag{3}$$

where the inner optimization can be approximated via the first order Taylor expansion, which results in the following solution ($\rho$ is a hyper-parameter):

$$\hat{\boldsymbol{\epsilon}}(\boldsymbol{\theta}) = \rho \nabla_{\boldsymbol{\theta}} \mathcal{L}(\boldsymbol{\theta}) / \left( \|\nabla_{\boldsymbol{\theta}} \mathcal{L}(\boldsymbol{\theta})\|_2^2 \right)^{\frac{1}{2}} \tag{4}$$

**Overall Optimization Procedure:** At the $t_{\text{th}}$ step of the SFE stage, we first compute the original loss $\mathcal{L}(\boldsymbol{\theta})$ for a mini-batch, and obtain the perturbative loss $\mathcal{L}_{SAM} = \mathcal{L}(\boldsymbol{\theta} + \hat{\boldsymbol{\epsilon}}(\boldsymbol{\theta}))$ according to Eqn. 4, as well as the variability collapse loss $\mathcal{L}_{vc}$ defined in Eqn. 2 based on perturbative loss. Thus we have the average gradient $\boldsymbol{g_0}$ and the class-specific gradient $\boldsymbol{g_i}, i \in \{1, 2, ..., K\}$ by back propagating $\mathcal{L}_{moo} = \mathcal{L}_{SAM} + \mathcal{L}_{vc}$. Finally, the dynamic class weights $\boldsymbol{\omega} = \{\omega_1, \omega_2, ..., \omega_K\}$ is obtained by solving CAGrad (Liu et al., 2021a):

$$\min_{\boldsymbol{\omega} \in \mathcal{W}} F(\omega) := \boldsymbol{g_{\omega}^{\top}} \boldsymbol{g_0} + \sqrt{\phi} \|\boldsymbol{g_{\omega}}\|, \text{ where } \phi = c^2 \|\boldsymbol{g_0}\|^2, \tag{5}$$

and update the model via: $\boldsymbol{\theta_t} = \boldsymbol{\theta_{t-1}} - \alpha \left( \boldsymbol{g_0} + \frac{\phi^{1/2}}{\|\boldsymbol{g_{\omega}}\|} \boldsymbol{g_{\omega}} \right), \boldsymbol{g_{\omega}} = \sum_i^K \omega_i * \boldsymbol{g_i}$. The overall pseudo algorithm is summarized in the **Section 6.2 of the Appendix**.

### 4.3 IMPLEMENTATION DETAILS

We implement our code with Python 3.8 and PyTorch 1.4.0, while all experiments are carried out on Tesla V100 GPUs. We train each model with batch size of 64 (for CIFAR10-LT and CIFAR100-LT) / 128 (for Places-LT) / 256 (for ImageNet-LT and iNaturalist), SGD optimizer with momentum of 0.9.

## 5 EVALUATION

Following the mainstream protocols, we conduct experiments on popular DLTR benchmarks: CIFAR10-/CIFAR100-LT, Places-LT (Liu et al., 2019), ImageNet-LT (Liu et al., 2019) and iNaturalist2018 (Van Horn et al., 2018). To show the versatility of PLOT, we equip it with the aforementioned popular re-balancing regimes under various imbalance scenarios. Moreover, PLOT achieves the state-of-the-art (SOTA) performance by augmenting the advanced baseline across large scale datasets. Micro benchmarks are elaborated to show the effectiveness of each components finally. For fair comparison, we exclude ensemble or pre-training models in our experiments.

Table 2: Performance on CIFAR datasets.

| | CIFAR10-LT | | | CIFAR100-LT | | |
|---|---|---|---|---|---|---|
| Imb. | 200 | 100 | 50 | 200 | 100 | 50 |
| LDAM-DRW | 71.38 | 77.71 | 81.78 | 36.50 | 41.40 | 46.61 |
| LDAM-DRW + PLOT | **74.32** | **78.19** | **82.09** | **37.31** | **42.31** | **47.04** |
| M2m | 73.43 | 77.55 | 80.94 | 35.81 | 40.77 | 45.73 |
| M2m + PLOT | **74.48** | **78.42** | **81.79** | **38.43** | **43.00** | **47.19** |
| cRT + Mixup | 73.06 | 79.15 | 84.21 | 41.73 | 45.12 | 50.86 |
| cRT + Mixup + PLOT | **78.99** | **80.55** | **84.58** | **43.80** | **47.59** | **51.43** |
| Logit Adjustment | - | 78.01 | - | - | 43.36 | - |
| Logit Adjustment + PLOT | - | **79.40** | - | - | **44.19** | - |
| MiSLAS | 76.59 | 81.33 | 85.23 | 42.97 | 47.37 | 51.42 |
| MiSLAS + PLOT | **77.73** | **81.88** | **85.70** | **44.28** | **47.91** | **52.66** |
| GCL | 79.25 | 82.85 | 86.00 | 44.88 | 48.95 | 52.85 |
| GCL + PLOT | **80.08** | **83.35** | **85.90** | **45.61** | **49.50** | **53.05** |

### 5.1 VERSATILITY VERIFICATION

Given the inherent optimization conflicts in advanced DLTR models, `PLOT` can serve as a valuable augmentation technique. Our experimental results, presented in Table 2, demonstrate that `PLOT` brings improvements in most scenarios by addressing the problem from a new dimension that is orthogonal to current solutions. Notably, cRT + Mixup and LDAM-DRW exhibit the most conflict scenarios and gain the most from `PLOT`. In fact, cRT + Mixup even achieves competitive performance compared to the state-of-the-art under certain imbalance ratio settings, highlighting the efficacy of `PLOT` in addressing optimization conflict problems. We also observe a marginal effect of this augmentation, as GCL exhibits marginal improvement or even degradation, owing to the absence of significant optimization conflicts (see Fig. 1).

## 5.2 Comparisons with SOTA

We further evaluate the effectiveness of `PLOT` on large-scale datasets, i.e., Places-LT, ImageNet-LT, and iNaturalist, and compare it against mainstream methods in Table 3. Through augmenting two advanced baselines (cRT + Mixup and MiS-LAS), `PLOT` achieves state-of-the-art performance. Specifically, our approach exhibits a substantial performance advantage over other DLTR models on Places-LT and iNaturalist, two recognized challenging benchmarks due to their high imbalance ratios.

## 5.3 Ablation Study

Our system comprises multiple components, and we aim to demonstrate their individual effectiveness. To this end, we conduct ablation studies and present the relationship between each component and the final performance in Table 4. Our results indicate that the proposed operations, i.e., temporal design (temp), variability collapse loss (var.) and anticipate worst-case optimization (anti.), can significantly enhance the system's performance.

## 5.4 Optimization Trajectory Analysis

We capture the optimization trajectories of different categories in LDAM-DRW + `PLOT` and present them in Fig. 8. In the left figure, the gradient similarity is computed between each category and the average gradient, while in the right figure, it is calculated between each category and the gradient aggregated by MOO. By comparison, the original LDAM-DRW approach exhibits a dominance of head classes in representation learning, resulting in a deterioration of shared feature extraction. In contrast, our augmented version with `PLOT` demonstrates relatively comparable and stable similarities among categories, indicating the potential for effective extraction of shared features across categories.

Table 3: Performance on large-scale Datasets.

| Dataset | Method | Backbone | Overall |
|---|---|---|---|
| Places-LT | CE | ResNet-152 | 30.2 |
| | Decouple-$\tau$-norm | ResNet-152 | 37.9 |
| | Balanced Softmax | ResNet-152 | 38.6 |
| | LADE | ResNet-152 | 38.8 |
| | RSG | ResNet-152 | 39.3 |
| | DisAlign | ResNet-152 | 39.3 |
| | ResLT | ResNet-152 | 39.8 |
| | GCL | ResNet-152 | 40.6 |
| | cRT + Mixup | ResNet-152 | 38.5 |
| | cRT + Mixup + `PLOT` | ResNet-152 | **41.0** |
| | MiSLAS | ResNet-152 | 40.2 |
| | MiSLAS + `PLOT` | ResNet-152 | 40.5 |
| ImageNet-LT | CE | ResNeXt-50 | 44.4 |
| | Decouple-$\tau$-norm | ResNet-50 | 46.7 |
| | Balanced Softmax | ResNeXt-50 | 52.3 |
| | LADE | ResNeXt-50 | 52.3 |
| | RSG | ResNeXt-50 | 51.8 |
| | DisAlign | ResNet-50 | 52.9 |
| | | ResNeXt-50 | 53.4 |
| | ResLT | ResNeXt-50 | 52.9 |
| | LDAM-DRW + SAM | ResNet-50 | 53.1 |
| | GCL | ResNet-50 | **54.9** |
| | cRT + Mixup | ResNet-50 | 51.7 |
| | cRT + Mixup + `PLOT` | ResNeXt-50 | 54.3 |
| | MiSLAS | ResNet-50 | 52.7 |
| | MiSLAS + `PLOT` | ResNet-50 | 53.5 |
| iNat-2018 | CE | ResNet-50 | 61.7 |
| | Decouple-$\tau$-norm | ResNet-50 | 65.6 |
| | Balanced Softmax | ResNet-50 | 70.6 |
| | LADE | ResNet-50 | 70.0 |
| | RSG | ResNet-50 | 70.3 |
| | DisAlign | ResNet-50 | 70.6 |
| | ResLT | ResNet-50 | 70.2 |
| | LDAM-DRW + SAM | ResNet-50 | 70.1 |
| | GCL | ResNet-50 | 72.0 |
| | cRT + Mixup | ResNet-50 | 69.5 |
| | cRT + Mixup + `PLOT` | ResNet-50 | 71.3 |
| | MiSLAS | ResNet-50 | 71.6 |
| | MiSLAS + `PLOT` | ResNet-50 | **72.1** |

Table 4: Ablation studies on CIFAR10-LT when imbalance ratio is set as 200.

| cRT + Mixup | w/ temp | w/ anti. | w/ var. | Acc. |
|---|---|---|---|---|
| ✔ | | | | 73.06 |
| ✔ | ✔ | | | 76.02 |
| ✔ | ✔ | ✔ | | 77.79 |
| ✔ | ✔ | ✔ | ✔ | **78.99** |

Figure 8: Gradient similarity with the aggregated gradient before / after applying `PLOT` on LDAM-DRW.

(a) Gradient similarities of LDAM-DRW. (b) Gradient similarities of `PLOT`.

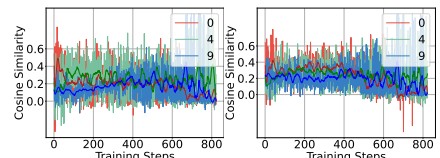

## 6 RELATED WORKS

### 6.1 DEEP LONG-TAILED RECOGNITION

Recent advancements in DLTR have been driven by three distinct design philosophies: (1) re-balancing strategies on various aspects, (2) ensemble learning, and (3) representation learning. In the first category, considerable efforts have been dedicated to re-sampling (Wang et al., 2019; Zang et al., 2021; Cai et al., 2021; Wei et al., 2021), re-weighting (Park et al., 2021; Kini et al., 2021), re-margining (Feng et al., 2021; Cao et al., 2019; Koltchinskii & Panchenko, 2002), logit adjustment (Menon et al., 2020; Zhang et al., 2021a), and information augmentation (Kim et al., 2020; Yang & Xu, 2020; Zang et al., 2021), etc. As anticipated, these approaches aim to manually re-balance the model by addressing sample number (through sampling and augmentation), cost sensitivity, prediction margin/logit, and other factors, thereby reducing the bias towards major classes. Ensemble learning-based approaches strive to leverage the expertise of multiple models. Generally, there are several methods for aggregating these models. BBN (Zhou et al., 2020) and SimCAL (Wang et al., 2020a) train experts using both long-tailed and uniformly distributed data and aggregate their outputs. On the other hand, ACE (Cai et al., 2021), ResLT (Cui et al., 2022), and BAGS (Li et al., 2020) train experts on different subsets of categories. SADE (Zhang et al., 2022) employs diverse experts, including long-tailed, uniform, and inverse long-tailed, and adaptively aggregates them using a self-supervised objective.

Recent efforts in representation learning have emerged in decoupling and contrastive learning, which employ distinct regimes to obtain general representations for all categories. Decoupling-based methods (Kang et al., 2019; Zhong et al., 2021; Li et al., 2022) have shown that the representation learned via random sampling strategies is powerful enough, and additional effort devoted to the second stage, i.e., classifier adjustment, can help achieve advanced performance. A recent study (Liu et al., 2021b) has empirically found that contrastive learning-based methods are less sensitive to imbalance scenarios. Thus, these methods (Cui et al., 2021; Zhu et al., 2022) extract general representations via supervised contrastive learning and achieve competitive performance. Our work takes a new approach to DLTR by developing a gradient conflict-averse solution, which is almost orthogonal to current solutions and has been verified to be effective.

### 6.2 MOO-BASED MTL

Multi-task learning (MTL), particularly MOO-based MTL, has garnered significant attention in the machine learning community as a fundamental and practical task. MGDA-UB (Sener & Koltun, 2018) achieves Pareto optimality by optimizing a derived upper bound in large-scale MTL scenarios. PCGrad (Yu et al., 2020) mitigates conflict challenges by projecting gradients on the corresponding orthogonal directions. In contrast, CAGrad (Liu et al., 2021a) develops a provably convergent solution by optimizing the worst relative individual task and constraining it around the average solution. Additionally, EPO (Mahapatra & Rajan, 2020) proposes a preference-guided method that can search for Pareto optimality tailored to the prior. Our work represents the first attempt to integrate MOO into DLTR. We bridge the gap between MTL and DLTR and propose two improvements for further augmentation. Although evaluating PLOT under the MTL setting would be a satisfactory choice, it is beyond the scope of this paper and is left for future investigation.

## 7 CONCLUSION

This paper have presented a novel approach to bridging the gap between MTL and DLTR. Specifically, we proposed a re-design of the MOO paradigm from structural to temporal, with the aim of addressing the challenge of optimization conflicts. To further ensure the convergence and generalization of the MOO algorithm, we optimized the derived MOO-based DLTR generation bound and seek a flatter minima. Our experimental results demonstrated the benefits of injecting the Pareto property across multiple benchmarks. We hope that our findings provide valuable insights for researchers studying the integration of MOO and DLTR, which has been shown to hold great promise. Our future works lie in developing adaptive strategies to apply MOO algorithm more efficiently.

## 8 ACKNOWLEDGEMENTS

We thank anonymous reviewers for their valuable comments. This work was supported by the NSFC under Grants 61932017 and 61971390.

## 9 REPRODUCIBILITY STATEMENT

Further implementation details can be found in **Section 2 of the Appendix**. Specifically, the supplementary material includes the attached code, which serves as a reference and provides additional information. The provided code demo consists of a prototype trained on the CIFAR10-/100-LT datasets.

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

# APPENDIX FOR: PARETO DEEP LONG-TAILED RECOGNITION: A CONFLICT-AVERSE SOLUTION

**Anonymous authors**

## CONTENTS

# 1 THEORETICAL PROOFS

## 1.1 PROOF OF THEOREM 4.1

**Theorem 1.** *(MOO-based DLTR Generalization Bound) If the loss function $l_k$ belonging to $k_{th}$ category is $M_k$-Lipschitz, and $\forall(\boldsymbol{x}, \boldsymbol{y}), (\boldsymbol{x'}, \boldsymbol{y'}) \in \mathcal{X} \times \mathcal{Y}, \forall \boldsymbol{h} \in \mathcal{H}$: $\|[\boldsymbol{h}(\boldsymbol{x}), \boldsymbol{y}] - [\boldsymbol{h}(\boldsymbol{x'}), \boldsymbol{y'}]\| \leq \mathcal{D}_{\mathcal{H}}$, assume $M_k \mathcal{D}_{\mathcal{H}}$ is bounded by $M$, then for any $\epsilon > 0$ and $\delta > 0$, with probability at least $1 - \delta$, the following inequality holds for $\forall \boldsymbol{h} \in \mathcal{H}$ and $\forall \boldsymbol{\omega} \in \mathcal{W}$:*

$$\mathcal{L}_{\boldsymbol{\omega}}(\boldsymbol{h}) \leq \hat{\mathcal{L}}_{\boldsymbol{\omega}}(\boldsymbol{h}) + 2\mathfrak{R}_S(\mathcal{G}, \boldsymbol{\omega}) + \sum_{k=1}^{K} \omega_k M \sqrt{\frac{\log \frac{1}{\delta}}{2}}$$

*Proof.* $\forall \boldsymbol{\omega} \in \mathcal{W}$ and sample $S = \{(\boldsymbol{x_1}, \boldsymbol{y_1}), ..., (\boldsymbol{x_N}, \boldsymbol{y_N})\}$, let $\Phi(S) = \sup_{\boldsymbol{h} \in \mathcal{H}} \mathcal{L}_{\boldsymbol{\omega}}(\boldsymbol{h}) - \hat{\mathcal{L}}_{\boldsymbol{\omega}}(\boldsymbol{h})$. And assume $S'$ is another sample set that contain only one point $(\boldsymbol{x'}, \boldsymbol{y'})$ different from $S$. Thus, we have

$$\begin{aligned}
\Phi(S') - \Phi(S) &= \sup_{\boldsymbol{h} \in \mathcal{H}}[\mathcal{L}_{\boldsymbol{\omega}}(\boldsymbol{h}) - \hat{\mathcal{L}}'_{\boldsymbol{\omega}}(\boldsymbol{h})] - \sup_{\boldsymbol{h} \in \mathcal{H}}[\mathcal{L}_{\boldsymbol{\omega}}(\boldsymbol{h}) - \hat{\mathcal{L}}_{\boldsymbol{\omega}}(\boldsymbol{h})] \\
&\leq \sup_{\boldsymbol{h} \in \mathcal{H}}[\mathcal{L}_{\omega}(\boldsymbol{h}) - \hat{\mathcal{L}}'_{\boldsymbol{\omega}}(\boldsymbol{h}) - \mathcal{L}_{\boldsymbol{\omega}}(\boldsymbol{h}) + \hat{\mathcal{L}}_{\boldsymbol{\omega}}(\boldsymbol{h})] \\
&= \sup_{\boldsymbol{h} \in \mathcal{H}}[\hat{\mathcal{L}}_{\boldsymbol{\omega}}(\boldsymbol{h}) - \hat{\mathcal{L}}'_{\boldsymbol{\omega}}(\boldsymbol{h})] \\
&= \sup_{\boldsymbol{h} \in \mathcal{H}} \left[ \sum_{k=1}^{K} \omega_k \frac{1}{m_k} \sum_{i=1}^{m_k} l(\boldsymbol{x'_i}, \boldsymbol{y'_i}) - \sum_{k=1}^{K} \omega_k \frac{1}{m_k} \sum_{i=1}^{m_k} l(\boldsymbol{x_i}, \boldsymbol{y_i}) \right] \\
&= \sup_{\boldsymbol{h} \in \mathcal{H}} \sum_{k=1}^{K} \omega_k \frac{1}{m_k} [l(\boldsymbol{x'_i}, \boldsymbol{y'_i}) - l(\boldsymbol{x_i}, \boldsymbol{y_i})] \\
&= \sup_{\boldsymbol{h} \in \mathcal{H}} \sum_{k=1}^{K} \omega_k \frac{1}{m_k} [l(\boldsymbol{x'_i}, \boldsymbol{y'_i}) - l(\boldsymbol{x_i}, \boldsymbol{y_i})] \\
&\leq \sup_{\boldsymbol{h} \in \mathcal{H}} \sum_{k=1}^{K} \omega_k \frac{1}{m_k} M_k \|[\boldsymbol{h}(\boldsymbol{x}), \boldsymbol{y}] - [\boldsymbol{h}(\boldsymbol{x'}), \boldsymbol{y'}]\| \\
&\leq \sup_{\boldsymbol{h} \in \mathcal{H}} \sum_{k=1}^{K} \omega_k \frac{1}{m_k} M_k \mathcal{D}_{\mathcal{H}} \\
&\leq \sup_{\boldsymbol{h} \in \mathcal{H}} \sum_{k=1}^{K} \omega_k M
\end{aligned}$$

According to McDiarmid's inequality, for any $\delta > 0$ with probability at least $1 - \delta$ for any $\boldsymbol{h} \in \mathcal{H}$, we have:

$$\begin{aligned}
\mathcal{L}_{\boldsymbol{\omega}}(\boldsymbol{h}) &\leq \hat{\mathcal{L}}_{\boldsymbol{\omega}}(\boldsymbol{h}) + \mathbb{E}\left[ \sup_{\boldsymbol{h} \in \mathcal{H}} \mathcal{L}_{\boldsymbol{\omega}}(\boldsymbol{h}) - \hat{\mathcal{L}}_{\boldsymbol{\omega}}(h) \right] + \sum_{k=1}^{K} \omega_k M \sqrt{\frac{\log \frac{1}{\delta}}{2}} \\
&\leq \hat{\mathcal{L}}_{\boldsymbol{\omega}}(\boldsymbol{h}) + 2\mathfrak{R}_S(\mathcal{G}, \boldsymbol{\omega}) + \sum_{k=1}^{K} \omega_k M \sqrt{\frac{\log \frac{1}{\delta}}{2}}
\end{aligned}$$

$\square$

## 1.2 PROOF OF THEOREM 4.2

**Pseudo Code of CAGrad:**

---

**Algorithm 1:** Training Paradigm of CAGrad

---

**Input:** Initial model parameter $\theta$, differentiable loss functions are $\mathcal{L}_i(\boldsymbol{\theta}), i \in [N]$
, a constant $c \in [0, 1]$ and learning rate $\alpha \in \mathbb{R}^+$.
**Output:** Model trained with CAGrad

**while** not converged **do**

At the $t_{th}$ optimization step, define $\boldsymbol{g_0} = \frac{1}{K} \sum_{i=1}^K \triangledown_{\boldsymbol{\theta}} \mathcal{L}_i(\boldsymbol{\theta_{t-1}})$ and $\phi = c^2 \|\boldsymbol{g_0}\|$.
Solve

$$\min_{\boldsymbol{\omega} \in \mathcal{W}} F(\omega) := \boldsymbol{g_\omega}^\top \boldsymbol{g_0} + \sqrt{\phi} \|\boldsymbol{g_\omega}\|, \text{ where } \phi = c^2 \|\boldsymbol{g_0}\|^2$$

Update $\boldsymbol{\theta_t} = \boldsymbol{\theta_{t-1}} - \alpha \left( \boldsymbol{g_0} + \frac{\phi^{1/2}}{\|\boldsymbol{g_\omega}\|} \boldsymbol{g_\omega} \right)$.

---

**Theorem 2.** *(Convergence of CAGrad in DLTR) With a fix step size $\alpha$ and the assumption of H-Lipschitz on gradients, i.e., $\|\triangledown \mathcal{L}_i(\boldsymbol{\theta}) - \triangledown \mathcal{L}_i(\boldsymbol{\theta}')\| \leq H \|\boldsymbol{\theta} - \boldsymbol{\theta}'\|$ for i = 1, 2, ..., K. Denote $\boldsymbol{d^*(\theta_t)}$ as the optimization direction of CAGrad at step t, then we have:*

$$\mathcal{L}(\boldsymbol{\theta_{t+1}}) - \mathcal{L}(\boldsymbol{\theta_t}) \leq -\frac{\alpha}{2}(1 - c^2) \|\boldsymbol{g_0(\theta_t)}\|^2 + \frac{\alpha}{2}(H\alpha - 1) \|\boldsymbol{d^*(\theta_t)}\|^2$$

*Proof.*

$$\mathcal{L}(\boldsymbol{\theta_{t+1}}) - \mathcal{L}(\boldsymbol{\theta_t}) = \mathcal{L}(\boldsymbol{\theta_t} - \alpha \boldsymbol{d^*(\theta_t)}) - \mathcal{L}(\boldsymbol{\theta_t})$$

$$\leq -\alpha \boldsymbol{g_0(\theta_t)}^\top \boldsymbol{d^*(\theta_t)} + \frac{H\alpha^2}{2} \|\boldsymbol{d^*(\theta_t)}\|^2$$

$$\leq -\frac{\alpha}{2} \left( \|\boldsymbol{g_0(\theta_t)}\|^2 + \|\boldsymbol{d^*(\theta_t)}\|^2 - \|\boldsymbol{g_0(\theta_t)} - \boldsymbol{d^*(\theta_t)}\|^2 \right) + \frac{H\alpha^2}{2} \|\boldsymbol{d^*(\theta_t)}\|^2$$

$$= -\frac{\alpha}{2} \left( \|\boldsymbol{g_0(\theta_t)}\|^2 - \|\boldsymbol{g_0(\theta_t)} - \boldsymbol{d^*(\theta_t)}\|^2 \right) + \frac{H\alpha^2}{2} \|\boldsymbol{d^*(\theta_t)}\|^2 - \frac{\alpha}{2} \|\boldsymbol{d^*(\theta_t)}\|^2$$

$$\leq -\frac{\alpha}{2}(1 - c^2) \|\boldsymbol{g_0(\theta_t)}\|^2 + \frac{\alpha}{2}(H\alpha - 1) \|\boldsymbol{d^*(\theta_t)}\|^2$$

$$\square$$

## 2 DETAILS OF IMPLEMENTATIONS

We conduct all experiments according to their publicly released code if applicable, please refer to these code for more details. Our early stop hyper-parameter $E$ is selected from $\{10, 30, 50, 80\}$, while anticipating worst-case optimization hyper-parameter $\rho$ is searched over $\{1.0e\text{-}3, 1.0e\text{-}4, 1.0e\text{-}5\}$.

### 2.1 DLTR METHODS

**LDAM-DRW.** LDAM-DRW re-balances the model via logit adjustment. It enforces the theoretical derived margins that is class frequency-related to achieve cost-sensitive learning. Its publicly released code can be found at `https://github.com/kaidic/LDAM-DRW`.

**Balanced Softmax.** Balanced Softmax is another cost-sensitive learning approach, which re-formulate softmax function in with the combination of link function and Bayesian inference. Its publicly released code can be found at `https://github.com/jiawei-ren/BalancedMetaSoftmax-Classification`.

**M2m.** M2m takes advantage of generating adversarial samples from major to minor classes and thus re-balance via augmentation. Its publicly released code can be found at `https://github.com/alinlab/M2m`.

**cRT + Mixup.** cRT is a milestone decoupling method, which re-trains the classifier with a balanced sampling strategy in the second stage. Despite the official implementation is available, it does not

include a version that utilizes ResNet-32 and is evaluated on CIFAR10-/CIFAR100-LT, which are the mainstream protocols. Therefore, we have re-implemented the method and achieved similar performance to that reported in GCL. In our implementation, we have adopted the same learning rate decay strategy as GCL, which involves multiplying the learning rate by 0.1 after the $160^{th}$ and $180^{th}$ epochs.

**MiSLAS.** MiSLAS follows the decoupling regime and adopts a class-frequency related label smoothing operation to achieve both the improvement of accuracy and calibration. Its publicly released code can be found at `https://github.com/dvlab-research/MiSLAS`.

**GCL.** Likewise, GCL is also a pioneer two-stage method, which observes the problem of softmax saturation and proposes to tackle it by Gaussian perturbation of different class logits with varied amplitude. Its publicly released code can be found at `https://github.com/Keke921/GCLLoss`.

**difficultyNet.** As a dynamic re-balancing method, difficultyNet employs meta learning to learn the adjustment of class re-weighting from logits. Its publicly released code can be found at `https://github.com/hitachi-rd-cv/Difficulty_Net`.

**BBN.** BBN Zhou et al. (2020) takes care of both representation learning and classifier learning by equipping with a novel cumulative learning strategy on two branches. Its publicly released code can be found at `https://github.com/megvii-research/BBN`.

**PaCo.** PaCo Cui et al. (2021) introduces a set of parametric class-wise learnable centers to re-balance from an optimization perspective, addressing the bias on high-frequency classes in supervised contrastive loss. Its publicly released code can be found at `https://github.com/dvlab-research/Parametric-Contrastive-Learning`.

**BCL.** BCL Zhu et al. (2022) proposes class-averaging and class-complement methods to help form a regular simplex for representation learning in supervised contrastive learning. Its publicly released code can be found at `https://github.com/FlamieZhu/Balanced-Contrastive-Learning`.

## 2.2 MOO-BASED MTL METHODS

**MGDA.** MGDA is a classical baseline for MOO-based MTL. This approach is particularly appealing due to its ability to guarantee convergence to a Pareto stationary point under mild conditions. Building upon this foundation, MGDA-UP introduces an upper bound on the multi-objective loss, aiming to optimize it and thereby achieve the Pareto optimal solution. In practice, it suggests task weighting based on the Frank-Wolfe algorithm (Jaggi, 2013). We conduct evaluations with the re-implementation in the publicly released code of CAGrad.

**EPO.** Different from the general MOO-based MTL, EPO provides a preference-specific MOO frameworks, which can effectively finds the expected Pareto front from the Pareto set by carefully controlling ascent to traverse the Pareto front in a principled manner. Generally, the re-balancing purpose also requires a preference for MOO. Its publicly released code can be found at `https://github.com/dbmptr/EPOSearch`.

**CAGrad.** CAGrad improves MGDA mainly by the ideas of worst-case optimization and convergence guarantee. It strikes a balance between Pareto optimality and globe convergence by regulating the combined gradients in proximity to the average gradient.:

$$\max_{\boldsymbol{d}\in\mathbb{R}^m}\min_{\boldsymbol{\omega}\in\mathcal{W}}\boldsymbol{g}_{\boldsymbol{\omega}}^{\top}\boldsymbol{d} \quad \text{s.t.} \|\boldsymbol{d}-\boldsymbol{g_0}\| \leq c\|\boldsymbol{g_0}\| \tag{1}$$

where $\boldsymbol{d}$ represents the combined gradient, while $\boldsymbol{g_0}$ denotes the averaged gradient, and $c$ is the hyperparameter. Its publicly released code can be found at `https://github.com/Cranial-XIX/CAGrad`.

**Why we choose CAGrad as the baseline?** CAGrad is widely recognized as a robust baseline in MOO -based MTL. In contrast to MGDA, which consistently favors individuals with smaller gradient norms, CAGrad achieves a delicate balance between Pareto optimality and global convergence. This unique characteristic allows CAGrad to preserve the Pareto property while maximizing individual progress. Conversely, EPO necessitates manual tuning of the preference hyper-parameter, which plays a crucial role in its performance but proves challenging to optimize in practical scenarios, particularly

for classification tasks with a large number of categories. In comparison, CAGrad requires less effort in terms of hyper-parameter tuning.

## 2.3 DATASETS

**CIFAR10-/CIFAR100-LT.** CIFAR10-/CIFAR100-LT is a subset of CIFAR10/CIFAR100, which is formed by sampling from the original 50,000 training images to create a long-tailed distribution. In our evaluation, we set the imbalance ratio $\beta = \frac{n_{max}}{n_{min}}$ to {200, 100, 50}, where $n_{max}$ and $n_{min}$ represent the sample numbers of the most and least frequent classes, respectively.

**Places-LT & ImageNet-LT.** Places-LT and ImageNet-LT are long-tailed variants of Places-365 and ImageNet, respectively. Places-LT comprises a total of 62.5K training images distributed across 365 classes, resulting in an imbalance ratio of 996. Similarly, ImageNet-LT consists of 115.8K training images spanning 1000 classes, with an imbalance ratio of 256.

**iNaturalist 2018.** iNaturalist 2018 is a naturally occurring long-tailed classification dataset that comprises 437.5K training images distributed across 8142 categories, resulting in an imbalance ratio of 512. In our evaluations, we have followed the official split.

## 3 OPEN LONG-TAILED RECOGNITION EVALUATION

To further demonstrate the robustness of the learned representations by `PLOT`, we conduct an evaluation known as open long-tailed recognition (OLTR) (Liu et al., 2019) on Places-LT and ImageNet-LT datasets. The results are presented in Table 1. Based on the comparison of F-measures, our method achieves state-of-the-art performance in OLTR. It is worth noting that we employ a cosine similarity measurement between incoming representations and prototypes proposed by (Liu et al., 2019), which allows us to compete favorably against bells and whistles OLTR methods (e.g., LUNA (Cai et al., 2022)). This highlights the superiority of the mechanism proposed by `PLOT` for representation learning.

Table 1: Open long-tailed recognition on Places-LT and ImagetNet-LT.

| F-measure ↑ | CE | Lifted Loss | Focal Loss | Range Loss | OpenMax | OLTR | IEM | LUNA | CC-SAM | cRT + Mixup + `PLOT` |
|---|---|---|---|---|---|---|---|---|---|---|
| ImageNet-LT | 0.295 | 0.374 | 0.371 | 0.373 | 0.368 | 0.474 | 0.525 | **0.579** | 0.552 | 0.563 |
| Places-LT | 0.366 | 0.459 | 0.453 | 0.457 | 0.458 | 0.464 | 0.486 | 0.491 | 0.510 | **0.516** |

## 4 ADDITIONAL EXPERIMENT RESULTS

### 4.1 IDENTIFY OPTIMIZATION CONFLICT UNDER VARIOUS CONDITIONS

Generally, the optimization objectives of samples from different classes tend to exhibit some conflicts, as they need to learn class-specific features in addition to the shared features across classes. In the case of a balanced training set, this conflict does not significantly impact the simultaneous optimization of individual classes. However, in an imbalanced scenario, the model optimization becomes dominated by the majority class, exacerbating this conflict issue. Consequently, the performance of the minority class is compromised in favor of optimizing the majority class. This outcome hinders the effective learning of category-sharing features. Importantly, this problem is intuitively independent of hyperparameters, optimizers, and other related factors.

To validate the aforementioned hypothesis, we have conducted an extensive investigation into the gradient conflict and dominated categories issues by thoroughly examining LDAM-DRW across various experimental settings. These settings encompassed diverse mini-batch sizes, learning rates, and optimizers. The meticulous presentation of our findings is shown in Fig. 1 and Fig. 2. Notably, Fig. 1 provides a snapshot of the instantaneous gradient conflict status at an early stage, while Fig. 2 illustrates the continuous gradient similarity status throughout the optimization process. It is important to emphasize that Figs. 2 (a) and (b) are presented on a larger scale in their axes, resulting in a less apparent discrepancy. However, it should be noted that these subfigures do not differ significantly from the other subfigures when placed on the same scale.

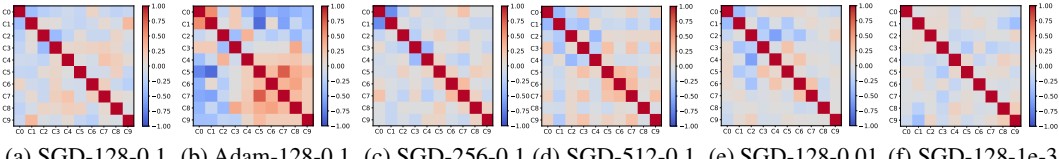

(a) SGD-128-0.1 (b) Adam-128-0.1 (c) SGD-256-0.1 (d) SGD-512-0.1 (e) SGD-128-0.01 (f) SGD-128-1e-3

Figure 1: Gradient conflict status of LDAM-DRW under various conditions. Each sub-figure is named as `(Optimizer)-(Batch-Size)-(Learning-Rate)`.

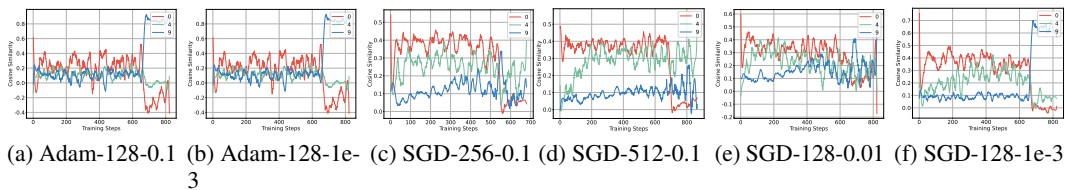

(a) Adam-128-0.1 (b) Adam-128-1e- (c) SGD-256-0.1 (d) SGD-512-0.1 (e) SGD-128-0.01 (f) SGD-128-1e-3
3

Figure 2: Continual gradient similarity status of LDAM-DRW under various conditions. Each sub-figure is named as `(Optimizer)-(Batch-Size)-(Learning-Rate)`.

As anticipated, we have observed the presence of the gradient conflict and dominated categories issues in all tested scenarios. These findings significantly contribute to validating the universality of the gradient conflict issue. If deemed applicable, we are fully committed to providing additional evidence and conducting further analysis in the updated version to reinforce the robustness of our findings.

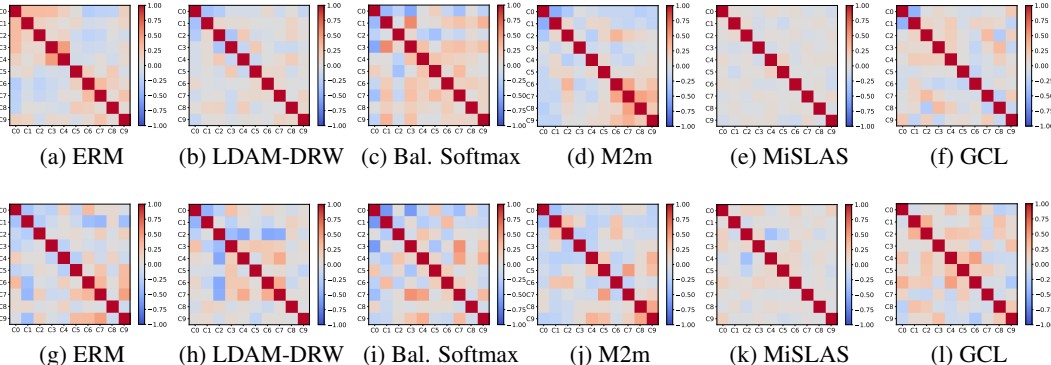

(a) ERM (b) LDAM-DRW (c) Bal. Softmax (d) M2m (e) MiSLAS (f) GCL

(g) ERM (h) LDAM-DRW (i) Bal. Softmax (j) M2m (k) MiSLAS (l) GCL

Figure 3: Gradient conflicts among categories at step 50 (a-f) and step 84(g-l).

## 4.2 GRADIENT CONFLICTS IN THE EARLY TRAINING STAGE

Here we further present the existing of optimization conflicts at different steps in Fig. 3. As depicted, such conflicts remain for a long time in the early stage.

## 4.3 FREQUENCY VS. LOSS VALUE

In Fig. 4, we present our observations of class-wise weighted loss, denoted as $\frac{B_k \cdot \bar{l}(\boldsymbol{x}_*^k, \boldsymbol{y}_*^k)}{B}$, to compare the norms of $B_*/B$ and $\bar{l}(\boldsymbol{x}_*^k, \boldsymbol{y}_*^k)$. Our results indicate that BalancedSoftmax, which functions as a re-balancing method by adjusting the loss function, is still dominated by class frequency, i.e., $B_*/B$. Conversely, LDAM-DRW exhibits a balanced weighted loss for both head and tail classes, thereby demonstrating the effectiveness of its re-balancing strategy.

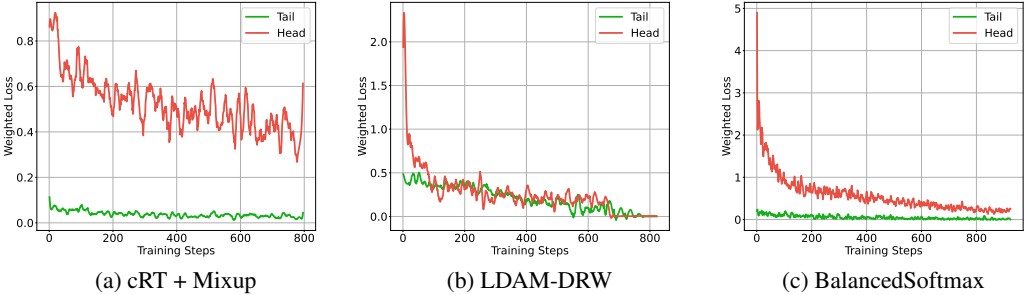

(a) cRT + Mixup         (b) LDAM-DRW        (c) BalancedSoftmax

Figure 4: Class-wise weighted loss comparison on CIFAR10-LT when imbalance ratio is set as 200 and batch size is set as 64.

## 4.4 GRADIENT CONFLICTS STATUS OF DYNAMIC RE-WEIGHTING APPROACH

As a form of dynamic re-weighting method, we further investigate the optimization conflicts associated with other dynamic re-weighting approaches to showcase our distinct research perspective. In this regard, we employ difficultyNet as the baseline for conducting a verification experiment, and the results are depicted in Fig. 5. It is evident that difficultyNet fails to address optimization conflicts during its early training stage, primarily due to its inherent lack of design nature.

Figure 5: Optimization conflicts of diff. Net.

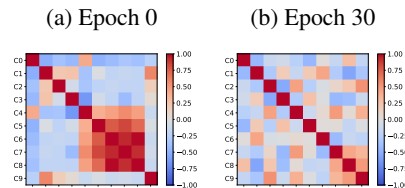

(a) Epoch 0       (b) Epoch 30

## 4.5 GRADIENT NORM EXAMINATION

In this section, we expand our analysis by conducting a thorough investigation into the phenomenon of gradient norm domination within mainstream DLTR approaches. To quantify this domination, we calculate the mean gradient of the corresponding category and present the results of our examination in Figure 6(a-f). As depicted, all the examined approaches demonstrate a consistent pattern, mirroring the tendency observed in Figure 2 of the main text. Specifically, we observe that the tail class is extremely under-optimized, indicating the explicit dominance of specific categories during the optimization process. These compelling findings provide strong support for the motivation underlying our proposed method, which aims to effectively mitigate such domination and its associated adverse effects. For comparative purposes, we also provide the `PLOT`-augmented gradient norm examinations in Figure 6(g-l). As expected, the results reveal that no individuals (categories) exhibit explicit domination during the early stage of representation learning.

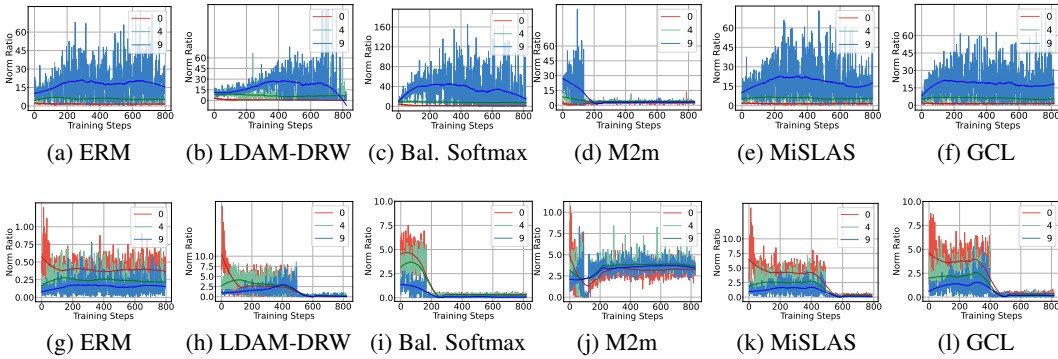

(a) ERM    (b) LDAM-DRW   (c) Bal. Softmax    (d) M2m      (e) MiSLAS      (f) GCL

(g) ERM    (h) LDAM-DRW    (i) Bal. Softmax    (j) M2m      (k) MiSLAS      (l) GCL

Figure 6: Gradient norm examination for mainstream DLTR approaches (a-f). 'Norm Ratio' is calculated between the individuals and the mean gradient. (g-l) are the `PLOT`-augmented ones.

### 4.6 GRADIENT SIMILARITY EXAMINATION AFTER PLOT AUGMENTATION

To substantiate the effectiveness of the `PLOT`, which facilitates simultaneous progress for all individuals, we compute the cosine similarities between individuals and the gradient derived from MOO. These cosine similarities are then compared with the corresponding vanilla results presented in Figure 2 of the main text. The comparative results are depicted in Figure 7. As observed, the DLTR methods augmented with `PLOT` exhibit a more balanced cosine similarity, indicating that all individuals achieve more equitable progress and improvements.

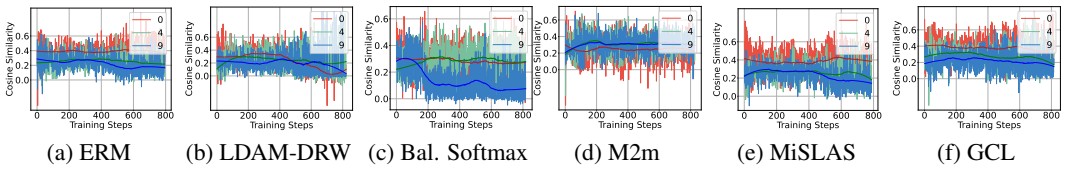

(a) ERM       (b) LDAM-DRW   (c) Bal. Softmax   (d) M2m       (e) MiSLAS      (f) GCL

Figure 7: Gradient similarity examination for `PLOT`-augmented mainstream DLTR approaches. 'Cosine Similarity' is calculated between the individuals and the MOO derived gradient.

### 4.7 APPLY PLOT ON DIFFERENT LAYERS

MOO algorithms are recognized to be computationally intensive, particularly when the number of tasks scales up. To address this issue, we propose applying `PLOT` to select specific layers rather than the entire model. In this study, we examine the impact of different layers to demonstrate the robustness of our approach. Specifically, we apply `PLOT` to different layers, i.e., layer1, layer2, and layer3 in ResNet-32, and present their final performance in Figure 8. Our results indicate that the last layer is the optimal choice across various evaluations.

Figure 8: Impact of different layers.

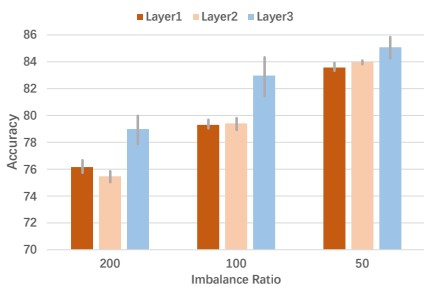

### 4.8 GRADIENT CONFLICTS STATUS UNDER THE BALANCED SETTING

We investigate the gradient conflict and similarity status in a balanced setting to motivate the integration of MOO and DLTR. To this end, we train ERM [1] with the full (roughly balanced) CIFAR10 dataset for 200 epochs and achieve a final accuracy of 92.84%. We record the class-wise gradient and compute their similarities. As shown in Fig 9, gradients among categories also exhibit conflict cases but play a roughly equal role during optimization [2].

## 5 LIMITATION

Figure 9: Gradient conflicts and similarity of ERM.

The limitations of `PLOT` can be attributed to two main factors. Firstly, the approach incurs additional computational overhead due to multiple gradient backpropagation and MOO. To address this issue, we propose applying `PLOT` to select layers rather than the entire model, which serves as an efficient alternative. Additionally, we randomly sample a small set of categories to participate in MOO for large-scale datasets. Secondly, the early stopping mechanism

(a) Gradient conflicts.    (b) Gradient sim.

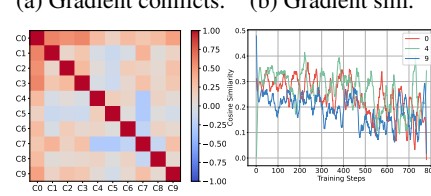

is currently controlled by a hyper-parameter, which introduces additional tuning complexities. To simplify this process, we select the hyper-parameter from a fixed set of values.

---

[1]We utilize WideResNets (Foret et al., 2020) as the backbone network without ShakeShake regularization or additional augmentations. Therefore, our reported result is slightly lower than the generally reported accuracy. Notably, our focus is on observing the class-wise gradient status rather than realizing a re-implementation.

[2]While this conclusion is straightforward, it is worth comparing with DLTR results.

# 6 COMPUTATIONAL COMPLEXITY AND SCALABILITY ANALYSIS

## 6.1 COMPUTATIONAL COMPLEXITY

As noted in the **Limitations** section, our method is more complex than its baselines. However, we would like to emphasize that our approach only applies MOO to a subset of the model parameters in a mini-batch, which contains fewer classes. This results in a lower time cost than expected. Additionally, we only apply MOO in the early stage of DLTR training for shared feature extraction. For large-scale datasets such as ImageNet-LT, we further adopt a class sampling strategy to reduce the computation cost. We believe that these optimizations help to mitigate the complexity of our approach while maintaining its effectiveness in addressing the long-tailed recognition problem. Here we provide a simple comparison of the time cost of running on a Tesla T4 GPU:

Table 2: Training time cost comparison running on Tesla T4.

| Method | CE | LDAM-DRW | cRT + Mixup | cRT + Mixup + `PLOT` | LDAM-DRW + `PLOT` |
|---|---|---|---|---|---|
| GPU Time (h) | 0.62 | 0.68 | 0.65 | 0.83 | 0.86 |

## 6.2 SCALABILITY

As previous research has indicated (Cadena et al., 2018; Hu et al., 2021; Zimmermann et al., 2021; Liu et al., 2019), the parameters of backbone layers tend to be more stable in the later stages of training. Therefore, our temporal-style MOO learning paradigm is designed to address the gradient conflict problem that occurs during the early stages of representation learning. This approach enables compatibility with mainstream DLTR models. Here, we provide examples of how our approach can be integrated with Decoupling using pseudo code, as shown in Algorithm 2 (Please refer Eqns. 2, 3, 4 in the maintext):

---

**Algorithm 2:** Representation Training Paradigm of Decoupling + `PLOT`

---

**Input:** Training Dataset $\mathcal{S} \sim p_s(\boldsymbol{x}, \boldsymbol{y}) = p_s(\boldsymbol{x}|\boldsymbol{y})p_s(\boldsymbol{y})$
**Output:** Model trained with `PLOT`

**Stage SFE**:
Initialize $\boldsymbol{\theta}$ randomly
**while** epoch $\leq$ E **do**
    **foreach** *batch* $\mathcal{B}_i$ *in* $\mathcal{S}$ **do**
        Compute empirical loss $L_{\mathcal{S}}$ with $\mathcal{B}_i$ and obtain its gradient $g_{\mathcal{S}}$
        Perturb $\boldsymbol{\theta}$ with $g_{\mathcal{S}}$ according to Eqn. 3 and Eqn. 4
        Compute the perturbative loss $\mathcal{L}_{SAM}$ and the variability collapse loss $\mathcal{L}_{vc}$ according to
          Eqn. 2.
        Estimate the class-specific gradient set $G = \{\boldsymbol{g_1}, \boldsymbol{g_2}, ..., \boldsymbol{g_k}\}$ with respect to
          $\mathcal{L}_{moo} = \mathcal{L}_{SAM} + \mathcal{L}_{vc}$
        Update $\boldsymbol{\theta}$ by solving Eqn. 6: $\boldsymbol{\theta} \leftarrow \boldsymbol{\theta} - \alpha \left( \boldsymbol{g_0} + \frac{\phi^{1/2}}{\|\boldsymbol{g_\omega}\|}\boldsymbol{g_\omega} \right)$.

**Stage TSO**:
**while** epoch $>$ E **do**
    **foreach** *batch* $\mathcal{B}_i$ *in* $\mathcal{S}$ **do**
        Computing empirical loss $L_{\mathcal{S}}$ with $\mathcal{B}_i$
        Update $\boldsymbol{\theta}$: $\boldsymbol{\theta} \leftarrow \boldsymbol{\theta} - \bigtriangledown_{\boldsymbol{\theta}}\mathcal{L}_{\mathcal{S}}(\boldsymbol{\theta})$

---

# 7 VISUALIZATIONS

## 7.1 FEATURE REPRESENTATION ANALYSIS

More class-specific Hessian spectral analysis [3] are provided in Figs. 10 11, which align with the conclusion in the main text, i.e., `PLOT` leads to a flatter minima for each class, thereby reducing the risk of getting stuck at saddle points.

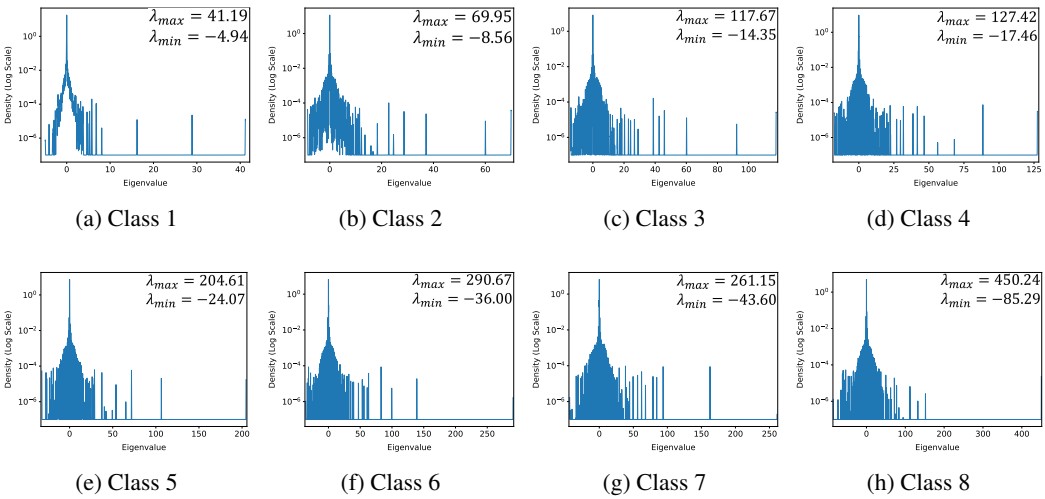

Figure 10: Hessian spectrum analysis of vanilla cRT + Mixup.

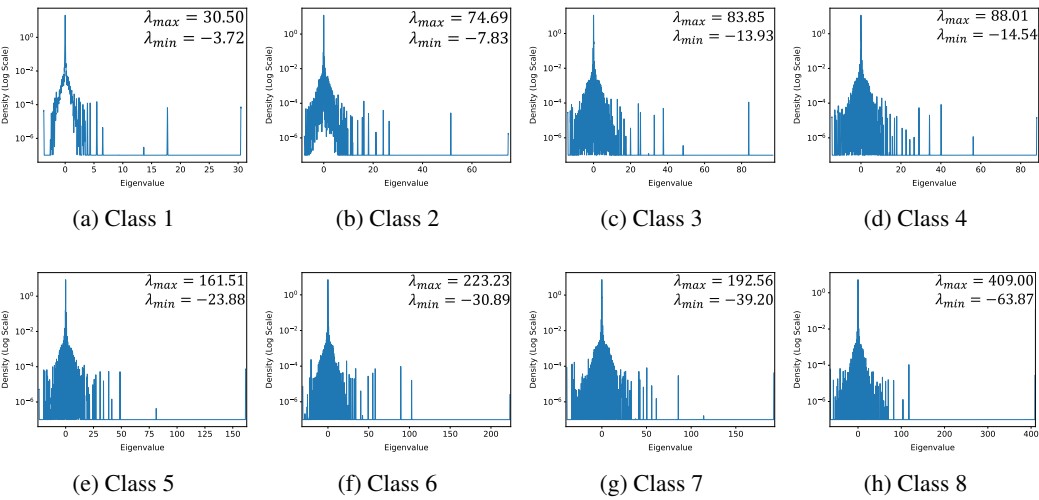

Figure 11: Hessian spectrum analysis of `PLOT`-augmented cRT + Mixup.

## 7.2 LOSS LANDSCAPE OF DLTR MODELS

To further illustrate the unattainable small value of $H$ in the DLTR models, we have included additional loss landscape visualizations in Fig. 12. These visualizations reveal that the tail classes of the models exhibit sharp minima, with the exception of M2m. Additionally, we observe that `PLOT` displays flat minima in both head and tail classes.

---

[3] https://github.com/val-iisc/Saddle-LongTail

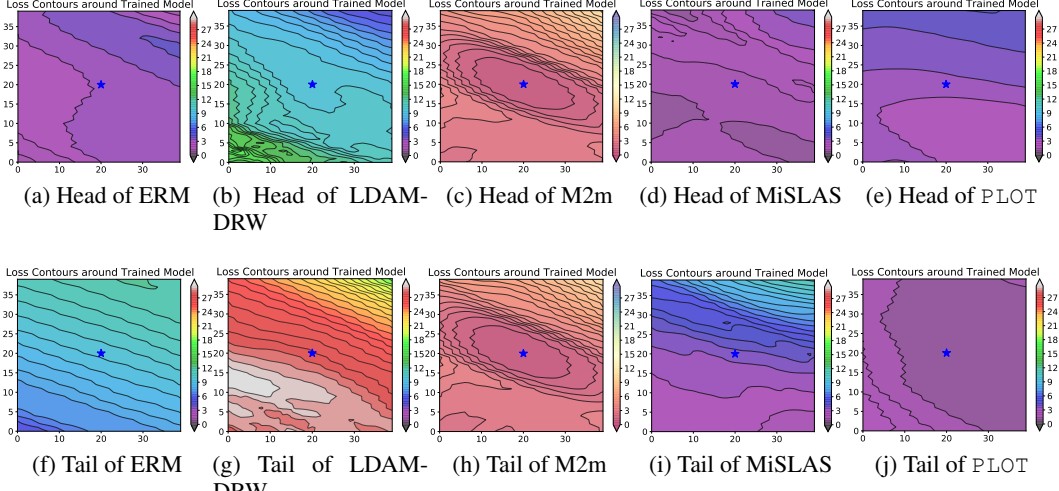

Figure 12: Loss landscapes of head and tail classes in ERM, LDAM-DRW, M2m, MiSLAS, and PLOT.

### 7.3 EMBEDDING VISUALIZATION

In addition, we present visualizations of the extracted embeddings from both the vanilla and PLOT-augmented approaches by projecting them into a 2D plane using t-SNE Van der Maaten & Hinton (2008). The corresponding visualizations can be found in Figure 13 (a)(b). As anticipated, the tail classes of the cRT + Mixup + PLOT approach exhibit increased separability compared to the vanilla approach. This observation suggests that the incorporation of PLOT enhances the representation of all categories, as intended.

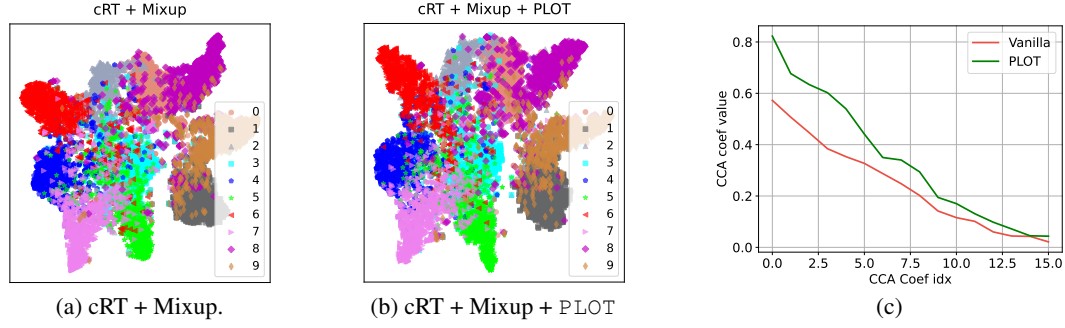

Figure 13: (a) and (b) are the t-SNE visualization results. (c) Representation similarity between head and tail classes.

### 7.4 CANONICAL CORRELATION ANALYSIS

To further investigate the shared feature representation across categories in the early stage, we examine the Canonical Correlation Analysis (CCA) scores (Raghu et al., 2017) of representations between head and tail categories in CIFAR10-LT, as learned by both the standard and PLOT enpowered LDAM-DRW. As illustrated in Fig. 13 (c), the PLOT augmented version exhibits a greater degree of similarity among categories in comparison to the conventional model, thereby substantiating the efficacy of our conflict-averse solution.

## 8 DETAILED RESULTS ON MAINSTREAM DATASETS

In order to demonstrate the generalization of PLOT and its impact on different subsets, we provide more detailed results in this section. Specifically, we compare PLOT with two state-of-the-art DLTR approaches, namely BBN and PaCo, by augmenting them with PLOT. The consistent improvements observed in Table 4 indicate that PLOT can effectively enhance various types of DLTR approaches. Furthermore, we present detailed results on large-scale datasets in Table 3, which empirically illustrate that PLOT successfully augments medium and tail classes without significantly compromising the performance of major classes.

| Dataset | Method | Backbone | Many | Medium | Few | Overall |
|---|---|---|---|---|---|---|
| Places-LT | CE | ResNet-152 | **45.7** | 27.3 | 8.20 | 30.2 |
| | Decouple-$\tau$-norm Kang et al. (2019) | ResNet-152 | 37.8 | 40.7 | 31.8 | 37.9 |
| | Balanced Softmax Ren et al. (2020) | ResNet-152 | 42.0 | 39.3 | 30.5 | 38.6 |
| | LADE Hong et al. (2021) | ResNet-152 | 42.8 | 39.0 | 31.2 | 38.8 |
| | RSG Wang et al. (2021) | ResNet-152 | 41.9 | 41.4 | 32.0 | 39.3 |
| | DisAlign Zhang et al. (2021) | ResNet-152 | 40.4 | 42.4 | 30.1 | 39.3 |
| | ResLT Cui et al. (2022) | ResNet-152 | 39.8 | **43.6** | 31.4 | 39.8 |
| | GCL Li et al. (2022) | ResNet-152 | - | - | - | 40.6 |
| | cRT + Mixup Kang et al. (2019) | ResNet-152 | 44.1 | 38.5 | 27.1 | 38.1 |
| | cRT + Mixup + PLOT | ResNet-152 | 40.9 | 43.1 | **37.1** | **41.0** |
| | MiSLAS Zhong et al. (2021) | ResNet-152 | 39.2 | 43.2 | 36.5 | 40.2 |
| | MiSLAS + PLOT | ResNet-152 | 39.2 | 43.5 | 36.9 | 40.5 |
| ImageNet-LT | CE | ResNeXt-50 | 65.9 | 37.5 | 7.7 | 44.4 |
| | Decouple-$\tau$-norm Kang et al. (2019) | ResNet-50 | 56.6 | 44.2 | 27.4 | 46.7 |
| | Balanced Softmax Ren et al. (2020) | ResNeXt-50 | 64.1 | 48.2 | 33.4 | 52.3 |
| | LADE Hong et al. (2021) | ResNeXt-50 | 64.4 | 47.7 | 34.3 | 52.3 |
| | RSG Wang et al. (2021) | ResNeXt-50 | 63.2 | 48.2 | 32.2 | 51.8 |
| | DisAlign Zhang et al. (2021) | ResNet-50 | 61.3 | 52.2 | 31.4 | 52.9 |
| | | ResNeXt-50 | 62.7 | 52.1 | 31.4 | 53.4 |
| | ResLT Cui et al. (2022) | ResNeXt-50 | 63.0 | 50.5 | 35.5 | 52.9 |
| | LDAM-DRW + SAM Rangwani et al. (2022) | ResNet-50 | 62.0 | 52.1 | 34.8 | 53.1 |
| | GCL Li et al. (2022) | ResNet-50 | - | - | - | 54.9 |
| | cRT + Mixup Kang et al. (2019) | ResNeXt-50 | 61.5 | 49.3 | 36.8 | 51.7 |
| | cRT + Mixup + PLOT | ResNeXt-50 | 62.6 | 52.0 | **39.4** | 54.3 |
| | MiSLAS Zhong et al. (2021) | ResNet-50 | 61.7 | 51.3 | 35.8 | 52.7 |
| | MiSLAS + PLOT | ResNet-50 | 61.4 | 52.3 | 37.5 | 53.5 |
| | BCL Zhu et al. (2022) | ResNet-50 | 65.7 | 53.7 | 37.3 | 56.0 |
| | BCL + PLOT | ResNet-50 | **67.4** | **54.4** | 38.8 | **57.2** |
| iNat-2018 | CE | ResNet-50 | 72.2 | 63.0 | 57.2 | 61.2 |
| | Decouple-$\tau$-norm Kang et al. (2019) | ResNet-50 | 65.6 | 65.3 | 65.9 | 65.6 |
| | Balanced Softmax Ren et al. (2020) | ResNet-50 | - | - | - | 70.6 |
| | LADE Hong et al. (2021) | ResNet-50 | - | - | - | 70.0 |
| | RSG Wang et al. (2021) | ResNet-50 | - | - | - | 70.3 |
| | DisAlign Zhang et al. (2021) | ResNet-50 | - | - | - | 70.6 |
| | ResLT Cui et al. (2022) | ResNet-50 | - | - | - | 70.2 |
| | LDAM-DRW + SAM Rangwani et al. (2022) | ResNet-50 | 64.1 | 70.5 | 71.2 | 70.1 |
| | GCL Li et al. (2022) | ResNet-50 | - | - | - | 72.0 |
| | cRT + Mixup Kang et al. (2019) | ResNet-50 | 74.2 | 71.1 | 68.2 | 70.2 |
| | cRT + Mixup + PLOT | ResNet-50 | **74.2** | 72.5 | 69.4 | 71.3 |
| | MiSLAS Zhong et al. (2021) | ResNet-50 | 73.2 | 72.4 | 70.4 | 71.6 |
| | MiSLAS + PLOT | ResNet-50 | 73.1 | **72.9** | **71.2** | **72.1** |

Table 3: Detailed Top-1 Accuracy on Places-LT, ImageNet-LT, and iNaturalist-2018.

Table 4: Top-1 Accuracy on CIFAR datasets.

| | CIFAR10-LT | | | CIFAR100-LT | | |
|---|---|---|---|---|---|---|
| Imb. | 200 | 100 | 50 | 200 | 100 | 50 |
| LDAM-DRW | 71.38 | 77.71 | 81.78 | 36.50 | 41.40 | 46.61 |
| LDAM-DRW + PLOT | **74.32** | **78.19** | **82.09** | **37.31** | **42.31** | **47.04** |
| M2m | 73.43 | 77.55 | 80.94 | 35.81 | 40.77 | 45.73 |
| M2m + PLOT | **74.48** | **78.42** | **81.79** | **38.43** | **43.00** | **47.19** |
| cRT + Mixup | 73.06 | 79.15 | 84.21 | 41.73 | 45.12 | 50.86 |
| cRT + Mixup + PLOT | **78.99** | **80.55** | **84.58** | **43.80** | **47.59** | **51.43** |
| Logit Adjustment | - | 78.01 | - | - | 43.36 | - |
| Logit Adjustment + PLOT | **-** | **79.40** | **-** | **-** | **44.19** | **-** |
| BBN | 73.52 | 77.43 | 80.19 | 36.14 | 39.77 | 45.64 |
| BBN + PLOT | **74.34** | **78.49** | **82.44** | **36.21** | **40.29** | **46.13** |
| MiSLAS | 76.59 | 81.33 | 85.23 | 42.97 | 47.37 | 51.42 |
| MiSLAS + PLOT | **77.73** | **81.88** | **85.70** | **44.28** | **47.91** | **52.66** |
| GCL | 79.25 | 82.85 | 86.00 | 44.88 | 48.95 | 52.85 |
| GCL + PLOT | **80.08** | **83.35** | 85.90 | **45.61** | **49.50** | **53.05** |
| PaCo | - | - | - | 47.28 | 51.71 | 55.74 |
| PaCo + PLOT | **-** | **-** | **-** | **47.75** | **52.60** | **56.61** |

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
