# OpenReview forum: "Pareto Deep Long-Tailed Recognition: A Conflict-Averse Solution"
_ICLR.cc/2024/Conference — ICLR 2024 poster_

### Official Review · Reviewer_wNAW · 2023-10-30

**Soundness:** 2 fair
**Presentation:** 3 good
**Contribution:** 2 fair
**Rating:** 6
**Confidence:** 4

**Summary:**

In this paper, the authors mainly tackle the long-tailed learning from the optimization perspective. Specially, this paper first reveals the optimization conflicts among categories in long-tailed learning, and proposes to integrate multi-objective optimization (MOO) with long-tailed learning. Temporal design on MOO, variability collapse loss and sharpness-aware minimization are then employed. Experiments on four long-tailed benchmark datasets are conducted to validate the effectiveness of the proposed method.

**Strengths:**

-	Leveraging MOO method to enhance long-tailed learning seems reasonable.
-	This paper reveals the phenomena of optimization conflicts among categories, which is an importance topic in the long-tailed learning from the optimization perspective.
-	The proposed method does not introduce additional computational cost.
-	The paper is generally easy to follow.

**Weaknesses:**

-	The paper introduces some trivial factors, e.g. sharpness-aware minimization, which makes the contribution of this paper ad-hoc. In my humble opinion, sharpness-aware minimization does not show strong connection to the proposed MOO framework, and can be integrated to almost all the long-tailed learning baselines.
-	The technical contribution of this paper seems not strong. This paper mainly adopts CAGRAD with slight modification on the training schedule (temporal design). Besides, sharpness-aware minimization is directly applied and the variability collapse loss is only a regularization term on the standard deviation of the loss.
-	The effectiveness of the proposed PLOT on top of baseline methods shows significant variability, see Table 1-2. However, the authors only conduct experiments on top of cRT + Mixup/MiSLAS on ImageNet-LT, Places-LT, iNaturalist, thus casting doubts on the confidence of the results.
-	In Table 1, MOO underperforms in nearly half the cases when compared with baseline methods (24/54). It cannot support the claims that MOO is beneficial for long-tailed learning. However, in Table 4, the authors present the most favorable performance enhancement with cRT + Mixup, thus making the results less confident.
-	According to Theorem 4.1, the generalization bound is bounded by the weighted intra-class loss variability, i.e., $w_KM$. It raises the question of whether uniformly constraining the intra-class loss variability is the optimal solution. Besides, in the neural collapse theory, minimizing loss function (e.g. cross-entropy loss) can lead to the intra-class collapse at the optimal case, why there should be an explicit constraint on intra-class variability?
-	This paper lacks background and discussions on the MOO methods, including MGDA, EPO and CAGrad. The reason for choosing CAGrad is solely performance-driven, lacking a connection to its inherent compatibility with the long-tailed learning paradigm.
-	In this paper, multi-classification task is regarded as multiple binary classification tasks. Does the proposed approach yield effective results when applied to imbalanced multi-label classification tasks? The empirical study on the multi-label tasks will benefit a lot to improve the generalizability of the proposed method.
-	This paper lacks disjoint analysis on the classification performance wr.t. different groups (many/medium/few).
-	Some results on logit adjustment are missing in Table 2.
-	There is no clue that the proposed method solves the gradient conflicts issue depicted in Figure 1.
-	In figure 8, it's unclear whether the proposed method actually surpass the performance of the baseline method LDAM-DRW in terms of gradient similarities. The gradient similarities of the proposed method exhibit a similar trend in the second stage.
-	It is suggested to provide some qualitative results (e.g. t-SNE) to validate the effectiveness of the proposed method on the embedding space.

**Questions:**

Please refer to the weakness part.

**Details Of Ethics Concerns:**

No.

---

> ### Author Response · Authors · 2023-11-15
> **Responses to Reviewer wNAW [1/3]**
>
> Thank Reviewer wNAW’s valuable time and comments and sorry for late response. To address your concerns, we provide pointwise responses below.
>
> > * Q1: The paper introduces some trivial factors, e.g. sharpness-aware minimization, which makes the contribution of this paper ad-hoc. In my humble opinion, sharpness-aware minimization does not show strong connection to the proposed MOO framework, and can be integrated to almost all the long-tailed learning baselines.
>
> A1: We acknowledge that SAM has been a technique commonly used to augment existing approaches. However, its effectiveness in the context of DLTR is not as significant as anticipated. While a previous study [1] demonstrated the effectiveness of incorporating SAM into certain DLTR methods, it was limited to a few baseline models (CE, LDAM, VS). However, recent research [2] indicates that SAM does not consistently improve a wide range of DLTR approaches. This suggests that the integration of SAM and DLTR requires a specific design rather than a straightforward application.
> In our paper, we employ SAM based on the observation that the convergence of CAGrad cannot be guaranteed in DLTR scenarios, particularly for tail classes that are known to exhibit sharp minima (larger H values). Although H-Lipschitz is typically assumed to hold, it fails to do so in DLTR scenarios and must be addressed. Therefore, we utilize SAM to address the challenges posed by integrating DLTR with MOO, rather than treating it as a mere technique or trick.
>
> Reference:
>
> [1] Harsh Rangwani, Sumukh K Aithal, Mayank Mishra, et al. Escaping saddle points for effective generalization on class-imbalanced data. Advances in Neural Information Processing Systems, 35: 22791–22805, 2022.
>
> [2] Zhipeng Zhou, Lanqing Li, Peilin Zhao, Pheng-Ann Heng, and Wei Gong. Class-conditional sharpness-aware minimization for deep long-tailed recognition. In Proceedings of the IEEE/CVF Conference on Computer Vision and Pattern Recognition (CVPR), pp. 3499–3509, June 2023.
>
> > * Q2: The technical contribution of this paper seems not strong. This paper mainly adopts CAGRAD with slight modification on the training schedule (temporal design). Besides, sharpness-aware minimization is directly applied and the variability collapse loss is only a regularization term on the standard deviation of the loss.
>
> A2:  We would like to emphasize that this paper makes significant contributions in three key aspects. Firstly, we provide a comprehensive observation that identifies an intrinsic conflict issue present in current DLTR approaches during the early stages of representation learning. This novel observation sheds light on an unexplored aspect of DLTR research.
> Secondly, we propose a framework, referred to as the temporal design, which enables the application of MOO to DLTR methods, effectively addressing the aforementioned conflict issue. This framework serves as a bridge between the distinct regimes of MOO-based MTL and DLTR. Furthermore, it holds potential for further enhancement through the incorporation of advanced MOO approaches. Lastly, we introduce two novel theoretical perspectives, i.e., SAM and variability collapse loss, to tackle the convergence and generalization challenges that arise when integrating DLTR with MOO. While these designs are not entirely novel, their purpose and re-discovery within the context of MOO-based DLTR scenarios are indeed novel and well motivated, as highlighted in the original main text.
>
> > * Q3: The effectiveness of the proposed PLOT on top of baseline methods shows significant variability, see Table 1-2. However, the authors only conduct experiments on top of cRT + Mixup/MiSLAS on ImageNet-LT, Places-LT, iNaturalist, thus casting doubts on the confidence of the results.
>
> A3: We acknowledge that the improvements in Table 1-2 show variability, which is in turns aligns with our conclusion, i.e., PLOT bring more improvements to those that have more serious conflict issue since a key contribution of PLOT is just to address the conflict issue for representation learning of DLTR. Besides, we have to clarify that most of the DLTR methods haven’t released their official implementations for large-scale datasets, making it hard for us to apply PLOT on them. But we would like to provide you some additional empirical evidence to demonstrate the effectiveness of PLOT on large-scale datasets in the next few days.

---

> > ### Author Response · Authors · 2023-11-15
> > **Responses to Reviewer wNAW [2/3]**
> >
> > > * Q4: In Table 1, MOO underperforms in nearly half the cases when compared with baseline methods (24/54). It cannot support the claims that MOO is beneficial for long-tailed learning. However, in Table 4, the authors present the most favorable performance enhancement with cRT + Mixup, thus making the results less confident.
> >
> > A4: We would like to clarify that our claim does not assert that all MOO methods are suitable for improving DLTR approaches. Even in the context of multi-task learning, many MOO approaches fail to surpass the performance of the naive linear scalarization strategy. However, our intention is to highlight that the application of MOO has the potential to benefit DLTR due to its inherent ability to address the conflict issue. If a well-selected or well-designed MOO approach is utilized, significant improvements can be achieved. The consistent and stable performance of CAGrad provides strong support for our claim. Consequently, we present an insightful framework that facilitates the integration of advanced MOO approaches to enhance DLTR. In our future work, we aim to provide additional integration results by employing more advanced MOO approaches.
> >
> > Furthermore, it is important to note that in Table 2 and Table 3, the PLOT method demonstrates substantial improvements for various DLTR methods, e.g., M2m on CIFAR100-LT. The cRT + Mixup approach, which serves as a baseline due to its simplicity, appears to yield greatest benefits in these tables.
> >
> > > * Q5: According to Theorem 4.1, the generalization bound is bounded by the weighted intra-class loss variability, i.e., $w_K M$. It raises the question of whether uniformly constraining the intra-class loss variability is the optimal solution. Besides, in the neural collapse theory, minimizing loss function (e.g. cross-entropy loss) can lead to the intra-class collapse at the optimal case, why there should be an explicit constraint on intra-class variability?
> >
> > A5: Although uniformly imposing intra-class loss variability may appear to offer an optimal solution, achieving it becomes challenging in imbalanced scenarios due to the unstable variability across different categories caused by significant differences in sample numbers. Consequently, we consider it solely as an auxiliary loss function.
> > Regarding the second question, we believe that recent studies [3,4,5] have addressed this concern. In [3], the issue of **Minority Collapse** under imbalanced scenarios is discussed, indicating that the Simplex ETF becomes unattainable as the tailed classes are compressed closer together. To address this problem, [4] proposed initializing the classifier as a random Simplex ETF, while [5] introduced a similar constraint to ours.
> >
> > Reference:
> >
> > [3] Cong Fang, Hangfeng He, Qi Long, and Weijie J Su. Exploring deep neural networks via layer-peeled model: Minority collapse in imbalanced training. Proceedings of the National Academy of Sciences, 118(43):e2103091118, 2021.
> >
> > [4] Yibo Yang, Shixiang Chen, Xiangtai Li, Liang Xie, Zhouchen Lin, and Dacheng Tao. Inducing neural collapse in imbalanced learning: Do we really need a learnable classifier at the end of deep neural network? Advances in Neural Information Processing Systems, 35:37991–38002, 2022.
> >
> > [5] Xuantong Liu, Jianfeng Zhang, Tianyang Hu, He Cao, Yuan Yao, and Lujia Pan. Inducing neural collapse in deep long-tailed learning. In International Conference on Artificial Intelligence and Statistics, pp. 11534–11544. PMLR, 2023.
> >
> > > * Q6: This paper lacks background and discussions on the MOO methods, including MGDA, EPO and CAGrad. The reason for choosing CAGrad is solely performance-driven, lacking a connection to its inherent compatibility with the long-tailed learning paradigm.
> >
> > A6: We have provided a section in **Section 2.2. of the Appendix** that introduces these MOO methods and as well as the reason that we choose CAGrad.
> > Why we choose CAGrad: CAGrad is widely recognized as a robust baseline in MOO -based MTL. In contrast to MGDA, which consistently favors individuals with smaller gradient norms, CAGrad achieves a delicate balance between Pareto optimality and global convergence. This unique characteristic allows CAGrad to preserve the Pareto property while maximizing individual progress. Conversely, EPO necessitates manual tuning of the preference hyper-parameter, which plays a crucial role in its performance but proves challenging to optimize in practical scenarios, particularly for classification tasks with a large number of categories. In comparison, CAGrad requires less effort in terms of hyper-parameter tuning.
> >
> > We will re-organized the paper to make sure these contents are involved in the main text.

---

> > > ### Author Response · Authors · 2023-11-15
> > > **Responses to Reviewer wNAW [3/3]**
> > >
> > > > * Q7: In this paper, multi-classification task is regarded as multiple binary classification tasks. Does the proposed approach yield effective results when applied to imbalanced multi-label classification tasks? The empirical study on the multi-label tasks will benefit a lot to improve the generalizability of the proposed method.
> > >
> > > A7: We acknowledge that extending the application of PLOT to multi-label tasks would significantly contribute to demonstrating its generalizability. However, conducting such an experiment is beyond the scope of this paper and is unfeasible within the constraints of our limited timeframe. Nevertheless, we are committed to providing supporting evidence for the corresponding results in our future research endeavors.
> > >
> > > > * Q8: This paper lacks disjoint analysis on the classification performance wr.t. different groups (many/medium/few).
> > >
> > > A8: Thanks for your reminder, due to the space limit of the main text, we now provide it in the **Section 8 of the Appendix**.
> > >
> > > > * Q9: Some results on logit adjustment are missing in Table 2.
> > >
> > > A9: We would like to clarify that the official code of logit adjustment (LA) solely offers implementation support for CIFAR10/100 datasets when the imbalance ratio is set to 100. It is important to note that the official code includes a specific pre-processing operation, making it infeasible to directly apply changes to the imbalance ratio. Therefore, for the sake of fair comparison, we only augment LA under the same conditions as specified in the official code.
> > >
> > > > * Q10: There is no clue that the proposed method solves the gradient conflicts issue depicted in Figure 1.
> > >
> > > A10: Please refer to gradient similarity examination in the **Section 7.2 of the Appendix**, and compare to those in Figure 2 in the main text. From the comparison, we can observe that PLOT augmented DLTR approaches show a more balanced similarities among individuals (categories), indicating the mitigating of conflict issue.
> > >
> > > > * Q11: In figure 8, it's unclear whether the proposed method actually surpass the performance of the baseline method LDAM-DRW in terms of gradient similarities. The gradient similarities of the proposed method exhibit a similar trend in the second stage.
> > >
> > > A11: We appreciate your concern. The observed similarity in the second stage can be attributed to the strong re-weighting strategy employed by LDAM-DRW, known as DRW. This strategy effectively enhances the dominance of the tail classes, leading to the observed trend.
> > >
> > > > * Q12: It is suggested to provide some qualitative results (e.g. t-SNE) to validate the effectiveness of the proposed method on the embedding space.
> > >
> > > A12: We have provided a t-SNE comparison between cRT + Mixup and cRT + Mixup + PLOT in the **Section 7.7 of the Appendix**. As anticipated, the tail classes of the cRT + Mixup + PLOT approach exhibit increased separability compared to the vanilla approach. This observation suggests that the incorporation of PLOT enhances the representation of all categories, as intended.

---

> > > > ### Author Response · Authors · 2023-11-19
> > > >
> > > > Dear Reviewer wNAW,
> > > >
> > > > We are reaching out to present additional empirical evidence that substantiates the generalization of our proposed method, PLOT. Specifically, we take a state-of-the-art contrastive learning based DLTR method called BCL [1] as the baseline and conduct experiments on ImageNet-LT. The ensuing results are presented below.
> > > >
> > > > | Method | Backbone | Many| Medium | Few| Overall |
> > > > | --- | --- | --- | --- | --- | --- |
> > > > | BCL | ResNet-50 | 65.7      | 53.7 | 37.3 | 56.0  |
> > > > | BCL+PLOT | ResNet-50 | **67.4**      | **54.4** | **38.8** | **57.2**  |
> > > >
> > > > As evident from the observations, PLOT continues to exhibit substantial improvements over BCL. Additionally, we extend the evaluation of PLOT to two other state-of-the-art approaches, i.e., BBN [2] and PaCo [3]. The subsequent results further validate the generalization capabilities of PLOT across diverse scenarios. It is important to note that all implementations are based on the official code provided by each respective method. To ensure a fair comparison, we exclusively present the results of PLOT augmentation on datasets for which the official code has been implemented. Moreover, both the vanilla and PLOT augmented versions are executed using the same three random seeds to maintain consistency.
> > > >
> > > > BBN on *CIFAR10-LT*:
> > > >
> > > > | Imb. Ratio| 200| 100 | 50|
> > > > | --- | --- | --- | --- |
> > > > | BBN        | 73.52      | 77.43 | 80.19 |
> > > > | BBN+PLOT   | **74.34**      | **78.49** | **82.44** |
> > > >
> > > > BBN and PaCo on *CIFAR100-LT*:
> > > >
> > > > | Imb. Ratio| 200| 100 | 50|
> > > > | --- | --- | --- | --- |
> > > > | BBN        | 36.14      | 39.77 | 45.64 |
> > > > | BBN+PLOT   | **36.21**     | **40.29** | **46.13** |
> > > > | PaCo| 47.28      | 51.71 | 55.74 |
> > > > | PaCo+PLOT   | **47.75**      | **52.60** | **56.61**|
> > > >
> > > > Reference:
> > > >
> > > > [1] Jianggang Zhu, Zheng Wang, Jingjing Chen, Yi-Ping Phoebe Chen, and Yu-Gang Jiang. Balanced contrastive learning for long-tailed visual recognition. InProceedings of the IEEE/CVF Conference on Computer Vision and Pattern Recognition, pp. 6908–6917, 2022.
> > > >
> > > > [2] Boyan Zhou, Quan Cui, Xiu-Shen Wei, and Zhao-Min Chen. Bbn: Bilateral-branch network with
> > > > cumulative learning for long-tailed visual recognition. InProceedings of the IEEE/CVF conference
> > > > on computer vision and pattern recognition, pp. 9719–9728, 2020.
> > > >
> > > > [3] Jiequan Cui, Shu Liu, Zhuotao Tian, Zhisheng Zhong, and Jiaya Jia. Reslt: Residual learning for
> > > > long-tailed recognition.IEEE Transactions on Pattern Analysis and Machine Intelligence, 2022.

---

> > ### Author Response · Authors · 2023-11-21
> >
> > Dear Reviewer wNAW,
> >
> > We sincerely thank you again for your great efforts in reviewing this paper. We have meticulously examined each of your points and taken utmost care in addressing them. As the deadline for the discussion period is drawing near, we would appreciate your feedback on whether our responses have adequately addressed your concerns.
> >
> > Best regards,
> >
> > Authors

---

> > > ### Comment · Reviewer_wNAW · 2023-11-22
> > >
> > > Thanks for the response. I have read the response and other reviews. Below are my follow-up comments.
> > >
> > > My concern is the technical contribution of this paper. After reading the rebuttal, I appreciate that the application of SAM and the variability collapse loss in long-tailed learning is reasonable. However, the introduction of both SAM and variability collapse loss, as they are already established methods, is directly utilized in this study without significant modification.
> > >
> > > > The observed similarity in the second stage can be attributed to the strong re-weighting strategy employed by LDAM-DRW, known as DRW. This strategy effectively enhances the dominance of the tail classes, leading to the observed trend.
> > >
> > > Does this mean that the proposed method achieves similar effect as the re-weighting strategy? If re-weighting can effectively address gradient conflicts in long-tailed situations, what then is the necessity of the proposed MOO framework?
> > >
> > > > T-SNE: As anticipated, the tail classes of the cRT + Mixup + PLOT approach exhibit increased separability compared to the vanilla approach.
> > >
> > > In Figure 15, it's unclear where the tail class is located, and whether the tail classes show improved separability compared to the baseline method.

---

> > > > ### Author Response · Authors · 2023-11-22
> > > > **Further Responses to Reviewer wNAW**
> > > >
> > > > We express our sincere gratitude for your additional feedback. We are pleased to note that we have managed to address the majority of your concerns. For the remaining issues, we have prepared detailed, point-by-point responses below.
> > > >
> > > > > Q1:   My concern is the technical contribution of this paper. After reading the rebuttal, I appreciate that the application of SAM and the variability collapse loss in long-tailed learning is reasonable. However, the introduction of both SAM and variability collapse loss, as they are already established methods, is directly utilized in this study without significant modification.
> > > >
> > > > A1: We acknowledge that in this paper, we have employed the SAM and variability collapse loss without significant modification. However, we wish to clarify that:
> > > > - The primary contribution of this paper lies in our novel approach to DLTR from a unique research perspective, specifically, the conflict issue in representation learning of DLTR. We provide substantial evidence to support our motivation and introduce a novel and scalable framework to integrate DLTR with MOO, which constitutes the main technical contribution of our paper.
> > > > - While the techniques of SAM and variability collapse loss are not novel per se, their application to the problems posed by this paper is well justified and not merely used as tricks. For instance, we provide a theoretical observation to derive the variability collapse loss from the perspective of MOO-based DLTR, which coincidentally shares the same form with neural collapse. Furthermore, SAM is applied based on our observations of the sharpness present in tail classes of current DLTR models. We acknowledge the significance of technical contribution, but we also maintain that it is equally important to justify the application of existing techniques to new problems with both theoretical and empirical evidence. For instance, [1] applies SAM to existing DLTR approaches without any modification, but it provides theoretical and empirical evidence to demonstrate the appropriateness of such an integration.
> > > >
> > > > Reference:
> > > >
> > > > [1] Harsh Rangwani, Sumukh K Aithal, Mayank Mishra, et al. Escaping saddle points for effective generalization on class-imbalanced data. Advances in Neural Information Processing Systems, 35: 22791–22805, 2022.
> > > >
> > > > >Q2: Does this mean that the proposed method achieves similar effect as the re-weighting strategy? If re-weighting can effectively address gradient conflicts in long-tailed situations, what then is the necessity of the proposed MOO framework?
> > > >
> > > > A2: It is important to note that we only apply MOO during the early stages of representation learning to preserve the features shared across all categories. However, the re-weighting strategy of LDAM, i.e., DRW, is indeed adopted at a later stage (after approximately 600 training steps), which is orthogonal to our method. Moreover, even with such a re-weighting strategy, the PLOT-augmented version still exhibits more balanced gradient similarities compared to the standard version, thereby demonstrating the effectiveness of PLOT. As a plug-and-play method, we also present an examination of gradient similarity with numerous other DLTR approaches (refer to **Figure 8 in Section 7.2 of the Appendix**), and generally, our methods all contribute to a more balanced gradient similarity across categories.
> > > >
> > > > >Q3: In Figure 15, it's unclear where the tail class is located, and whether the tail classes show improved separability compared to the baseline method.
> > > >
> > > > A3: The issue you are experiencing may be due to viewing the PDF through a web browser. We recommend utilizing local PDF software, such as Adobe Reader, for a more optimal viewing experience. We hope this suggestion resolves your problem.

---

> > > > > ### Comment · Reviewer_wNAW · 2023-11-22
> > > > >
> > > > > Thanks for the response. Most of my concerns have been addressed.  However, it is suggested to incorporate more comprehensive discussions on SAM-based long-tailed learning and neural collapse literature for better clarity of the proposed method. For example, "observations of the sharpness present in tail classes" have already been discovered. Additionally, exploring further discussions or potential extensions in multi-label long-tailed learning appears to be a promising direction.
> > > > >
> > > > > Overall, I appreciate the detailed empirical evidence (e.g., the gradient conflict) and the perspective of multi-objective optimization, which seems to be critical for long-tailed learning. I would like to recommend acceptance and raise my score to '6'.

---

> > > > > > ### Author Response · Authors · 2023-11-22
> > > > > > **Thank you for increasing your score!**
> > > > > >
> > > > > > We express our gratitude for your revised rating. We acknowledge the necessity for more comprehensive discussions on SAM-based long-tailed learning and neural collapse literature. We intend to incorporate these contents and investigate the potential application of our method to multi-label long-tailed learning problems in our future version. We extend our sincere thanks for your insightful discussion.

---

### Official Review · Reviewer_KQKE · 2023-11-03

**Soundness:** 3 good
**Presentation:** 3 good
**Contribution:** 3 good
**Rating:** 6
**Confidence:** 4

**Summary:**

This paper considers an interesting and important issue in long-tailed learning, that is the optimization conflicts. To solve this problem, the authors introduce Pareto optimization for long-tailed learning. First, the authors observe that existing widely-used fixed re-balancing strategies can uniformly lead to gradient conflict. Then, they introduce Multi-Objective Optimization (MOO) to enhance existing long-tailed learning methods. Moreover, the authors find that directly integrating MOO-based methods can lead to performance degradation. To solve this, they propose to decouple the MOO-based methods from the temporal rather than structural perspective to enhance the integration. Experimental results demonstrate that the proposed temporal MOO method can boost the baseline method and achieve an improvement by a large gap.

**Strengths:**

1. This paper studies an interesting problem, which is the optimization conflict in long-tailed learning.
2. The authors propose to utilize MOO to solve the conflict issue, which is reasonable and makes sense.
3. This paper is clearly written and easy to understand.
4. The authors conduct multiple empirical studies to demonstrate the effectiveness since the results show that the proposed method can boost the performance of the baseline methods obviously.
5. The source code is released for reproduction.

**Weaknesses:**

1. Figure 1 may cause misunderstandings. How is Figure 1 computed? Is it cosine similarities between the mean of the gradient of different classes? Since the diagonal is red (1.0), it seems that the values represent gradient similarities. And what is the connection between "gradient conflicts" and "gradient similarities"?
2. You mainly consider the directions of gradients. Have you considered the impacts of the L2 norms of the gradients?
3. As you have mentioned MOO is applied during the early stages of representation learning, how to select the applied stage?
4. The iNaturalist dataset has multiple versions. You should highlight it with "iNaturalist 2018".

**Questions:**

(This is not a question, but a suggestion). I found a very similar work that also studies long-tailed learning with multi-objective optimization [1]. Maybe you can publicly release your work considering its timeliness.

[1] Long-Tailed Learning as Multi-Objective Optimization, in https://arxiv.org/abs/2310.20490

---

> ### Author Response · Authors · 2023-11-11
> **Responses to Reviewer KQKE**
>
> We are glad that Reviewer KQKE recognize our paper to be interesting and easy to follow. To address Reviewer KQKE’s concerns, we provide pointwise responses below.
>
> > * Q1: Figure 1 may cause misunderstandings. How is Figure 1 computed? Is it cosine similarities between the mean of the gradient of different classes? Since the diagonal is red (1.0), it seems that the values represent gradient similarities. And what is the connection between "gradient conflicts" and "gradient similarities"?
>
> A1: Sorry for unclear description. The Figure 1 in the main text represents the cosine similarities among different classes, with the values indicating the degree of gradient similarity. As defined in **Definition 2.1**, when the cosine similarity between two gradients is less than 0, we classify them as conflicting, which is consistent with the definition in multi-task learning.
>
> > * Q2: You mainly consider the directions of gradients. Have you considered the impacts of the L2 norms of the gradients?
>
> A2: We acknowledge your concern regarding the consideration of both direction and norm in the context of our study. As depicted in **Figure 3(b)** in the main text, an increasing gradient imbalance exacerbates the issue of dominated conflicting, while non-conflicting scenarios also tend to bias the mean gradient towards the dominated one. To assess this imbalance issue, we calculate the cosine similarity between the mean gradient and the gradients of different classes, as presented in **Figure 2** in the main text. This analysis provides evidence that mainstream deep long-tailed approaches still exhibit varying degrees of imbalance issues from the perspective of gradient norm. Furthermore, we provide statistical evidence of dominated conflicts in Figure 5 in the main text to further substantiate our findings.
>
> > * Q3: As you have mentioned MOO is applied during the early stages of representation learning, how to select the applied stage?
>
> A3: The selection of the applied stage is determined by a hyper-parameter *E*; however, it is important to note that the selection is limited to a fixed set. For instance, if the long-tailed model is intended to be trained over a total of 200 epochs, we choose the applied stage *E* from the set {50, 80, 120, 140}. As part of our future directions, we aim to develop a strategy that can dynamically determine the applied stage, allowing for greater adaptability and optimization in the training process.
>
> > * Q4: The iNaturalist dataset has multiple versions. You should highlight it with "iNaturalist 2018".
>
> A4: We thank for your kind reminder, we will fix it in our updated version.
>
> > * Suggestion: I found a very similar work that also studies long-tailed learning with multi-objective optimization [1]. Maybe you can publicly release your work considering its timeliness.
>
> A5: We sincerely appreciate your reminder. Upon further investigation, we discovered that this paper was recently published on arXiv on October 31st. It is noteworthy that both our work and this paper share a common objective of integrating MOO and DLTR. However, it is important to highlight that our approaches follow distinct methodologies and regimes. Nonetheless, we genuinely value and appreciate your suggestion.

---

> > ### Comment · Reviewer_KQKE · 2023-11-12
> >
> > Thanks for your responses.
> >
> > - I suggest you update Figure 1 to avoid misunderstanding. For example, you can update the caption with more explanations.
> >
> > - The selection of the applied stage might be a weakness. More general approaches are needed for future work.
> >
> > - It seems that Figure 2 is irrelevant to the norms of gradients. Figure 2 only calculates the cosine similarity. Anyway, I believe that gradient conflicts exist in long-tailed learning w.r.t cosine similarity. However, a low cosine similarity can not necessarily indicate that the class is not optimized (maybe a negative cosine similarity can). The class might still be optimized well, with the gradient norm becoming small, even if the cosine similarity maintains a high level. So we need to take a look at the gradient norms of different classes during the whole training period.

---

> > > ### Author Response · Authors · 2023-11-13
> > >
> > > We appreciate your active response. Still, we would like to provide pointwise responses below.
> > >
> > > > * Q1: I suggest you update Figure 1 to avoid misunderstanding. For example, you can update the caption with more explanations
> > >
> > > A1: Thank you for your advice, we have fixed it in our updated version.
> > >
> > > > * Q2: The selection of the applied stage might be a weakness. More general approaches are needed for future work.
> > >
> > > A2: While we acknowledge the potential for improvement through the implementation of adaptive strategies, the primary objective of this paper is to present an effective framework for augmenting DLTR with MOO within the context of this traditional topic. However, we recognize the need for further refinement in our future work.
> > >
> > > > * Q3: It seems that Figure 2 is irrelevant to the norms of gradients. Figure 2 only calculates the cosine similarity. Anyway, I believe that gradient conflicts exist in long-tailed learning w.r.t cosine similarity. However, a low cosine similarity can not necessarily indicate that the class is not optimized (maybe a negative cosine similarity can). The class might still be optimized well, with the gradient norm becoming small, even if the cosine similarity maintains a high level. So we need to take a look at the gradient norms of different classes during the whole training period.
> > >
> > > A3: Figure 2 demonstrates the combined impact of gradient cosine similarities and norms. To provide further clarification, we have conducted a gradient norm examination similar to that depicted in Figure 2, as suggested. For detailed results, please refer to **Section 7.1 of the updated Appendix**. As depicted, the examination exhibits a similar tendency to Figure 2, indicating the domination of specific categories. The primary objective of this paper is to prevent such domination during the early stages of representation learning, aligning with the goals of MOO-based MTL approaches. In MOO-based MTL approaches [1][2][3], the focus is typically on promoting more balanced optimization by addressing both conflict and norm imbalance issues. The conflict issue often necessitates sacrificing individual progress, while the imbalance issue leads to uneven advancements among individuals. Therefore, through the examination of gradient similarities and norm ratios, as well as the amplification of dominated gradient conflicts (negative cosine similarity) under long-tailed scenarios in Figure 5, we provide a strong motivation for our approach. Furthermore, we also present the gradient norm examinations augmented with the *PLOT* for comparative purposes. As anticipated, the results reveal that no individuals exhibit explicit domination during the early stage of representation learning.
> > >
> > > Reference:
> > >
> > > [1] Aviv Navon, Aviv Shamsian, Idan Achituve, Haggai Maron,Kenji Kawaguchi, Gal Chechik, and Ethan Fetaya. Multi-task learning as a bargaining game. InInternational Conferenceon Machine Learning, pages 16428–16446. PMLR, 2022.
> > >
> > > [2] Dmitry Senushkin, Nikolay Patakin, Arseny Kuznetsov, andAnton Konushin. Independent component alignment for multi-task learning. InProceedings of the IEEE/CVF Conferenceon Computer Vision and Pattern Recognition, pages 20083–20093, 2023.
> > >
> > > [3] Bo Liu, Yihao Feng, Peter Stone, and Qiang Liu. Famo: Fast adaptive multi-task optimization. arXiv preprint arXiv:2306.03792, 2023.

---

> > > > ### Comment · Reviewer_KQKE · 2023-11-14
> > > >
> > > > Thank you for providing further responses.
> > > >
> > > > I think the motivation of the paper is a crucial highlight, so I suggest you enhance your insights regarding the gradient norm, and your response makes sense to me.
> > > >
> > > > I would like to recommend the acceptance. However, I also agree with the other reviewers' concerns, such as integrating PLOT with more state-of-the-art methods. I suggest the authors address these concerns.

---

### Official Review · Reviewer_yXjC · 2023-11-04

**Soundness:** 3 good
**Presentation:** 3 good
**Contribution:** 3 good
**Rating:** 6
**Confidence:** 5

**Summary:**

This paper focuses on the problem of Deep Long-Tailed Recognition (DTLR) and highlights the importance of dynamic re-balancing to address optimization conflicts in this domain. The authors empirically demonstrate that existing DTLR methods are dominated by certain categories due to fixed re-balancing strategies, preventing them from effectively handling gradient conflicts. To address this, they introduce an approach based on multi-objective optimization (MOO) to decouple the problem from a temporal perspective, avoiding class-specific feature degradation. Their method, named PLOT (Pareto deep long-tailed recognition), was conducted on several benchmarks for evaluation.

**Strengths:**

- The long-tailed recognition problem studied in this paper is a fundamental task that deserves further study.
- The paper is well organized and easy to follow.
- The technique contributions of this paper are novel and reasonable, which are tailored for the challenges of long-tailed recognition.

**Weaknesses:**

- My main concern with this paper is that the authors primarily demonstrate the effectiveness of their approach by augmenting existing methods with PLOT. However, they lack comprehensive performance comparisons by integrating PLOT with state-of-the-art methods like BBN, SADE, PaCo, etc. This omission makes it challenging to assess the method's effectiveness in comparison to the latest advancements in the field, raising questions about its overall impact and general applicability.
- Another potential weakness of this paper is the insufficient coverage of long-tailed recognition methods in the references. The paper may not thoroughly discuss some classical methods in the field, which can be essential for providing a comprehensive understanding of the long-tailed recognition landscape.
- The conclusion of the article should be written in the past tense. Additionally, the conclusions require to be added future work of this paper.
- The authors are encouraged to carefully proofread the paper.

**Questions:**

Please see the paper weaknesses.

---

> ### Author Response · Authors · 2023-11-15
> **Responses to Reviewer yXjC**
>
> Thank Reviewer yXjC’s valuable time and comments and sorry for late response. To address your questions, we provide pointwise responses below.
> > * Q1: My main concern with this paper is that the authors primarily demonstrate the effectiveness of their approach by augmenting existing methods with PLOT. However, they lack comprehensive performance comparisons by integrating PLOT with state-of-the-art methods like BBN, SADE, PaCo, etc. This omission makes it challenging to assess the method's effectiveness in comparison to the latest advancements in the field, raising questions about its overall impact and general applicability.
>
> A1: We appreciate your valuable advice. Following your suggestion, we have incorporated an advanced long-tailed method, i.e., BBN, to further showcase the effectiveness of PLOT on CIFAR10-/100-LT datasets. And the results are presented as below.
>
> CIFAR10-LT:
> | imb. ratio | 200        | 100   | 50 |
> | ---------- | ---------- | ------- | ----- |
> | BBN        | 73.52      | 77.43 | 80.19 |
> | BBN+PLOT   | 74.34 | 78.49 | 82.44 |
>
> CIFAR100-LT:
> | imb. ratio |  200         | 100   | 50|
> | ---------- | ------------- | ----- | ------- |
> | BBN        | 36.14       | 39.77 | 45.64 |
> | BBN+PLOT | 36.21       |40.29 | 46.13 |
>
> As observed, PLOT still benefits BBN under various imbalance ratio scenarios.
> It should be noted that all these implementations are based on their official implementations. For fair comparison, we only show the results of PLOT augmentation on those datasets that the official code has implemented on, and both the vanilla and PLOT augmented version are run on the same three random seeds.
> From above results, we believe the generalization of PLOT is well demonstrated.
>
> > * Q2:  Another potential weakness of this paper is the insufficient coverage of long-tailed recognition methods in the references. The paper may not thoroughly discuss some classical methods in the field, which can be essential for providing a comprehensive understanding of the long-tailed recognition landscape.
>
> A2: As suggested, we have additional introduced more related works involving re-balancing- and ensemble experts-based works. Please refer to our updated version.
>
> > * Q3: The conclusion of the article should be written in the past tense. Additionally, the conclusions require to be added future work of this paper.
>
> A3: Thank you for your reminder, we have fixed it.
>
> > * Q4: The authors are encouraged to carefully proofread the paper.
>
> A4: Thank you for your reminder. We are diligently conducting a thorough proofreading of the entire paper. If you have any further suggestions or advice regarding the writing, we kindly request you to bring them to our attention.

---

> > ### Author Response · Authors · 2023-11-19
> >
> > Dear Reviewer yXjC,
> >
> > We are reaching out to present additional empirical evidence that substantiates the generalization of our proposed method, PLOT. In accordance with your suggestion, we have incorporated two other state-of-the-art DLTR methods, i.e., BBN and PaCo, as baselines and augmented them with PLOT. It is worth noting that PaCo comprises both query and key branches, and we have exclusively applied PLOT to the query branch, as the key branch undergoes momentum updates from the query branch. The corresponding results are presented as follows.
> >
> > BBN on *CIFAR10-LT*:
> >
> > | Imb. Ratio| 200| 100 | 50|
> > | --- | --- | --- | --- |
> > | BBN        | 73.52      | 77.43 | 80.19 |
> > | BBN+PLOT   | **74.34**      | **78.49** | **82.44** |
> >
> > BBN and PaCo on *CIFAR100-LT*:
> >
> > | Imb. Ratio| 200| 100 | 50|
> > | --- | --- | --- | --- |
> > | BBN        | 36.14      | 39.77 | 45.64 |
> > | BBN+PLOT   | **36.21**     | **40.29** | **46.13** |
> > | PaCo| 47.28      | 51.71 | 55.74 |
> > | PaCo+PLOT   | **47.75**      | **52.60** | **56.61** |
> >
> > Due to the presence of two branches in PaCo, significant computational overhead is incurred, particularly when operating on large-scale datasets. Consequently, we have opted to evaluate an alternative state-of-the-art contrastive learning based DLTR method called BCL [1], which offers improved computational efficiency. In this context, we present the results of BCL on the *ImageNet-LT* dataset:
> >
> > | Method | Backbone | Many| Medium | Few| Overall |
> > | --- | --- | --- | --- | --- | --- |
> > | BCL | ResNet-50 | 65.7      | 53.7 | 37.3 | 56.0  |
> > | BCL+PLOT | ResNet-50 | **67.4**      | **54.4** | **38.8** | **57.2**  |
> >
> > Based on the above results, it is evident that PLOT exhibits significant improvements over its baselines across multiple benchmarks and DLTR regimes. This observation underscores the strong generalization capabilities of PLOT in a comprehensive manner.
> >
> > Reference:
> >
> > [1] Jianggang Zhu, Zheng Wang, Jingjing Chen, Yi-Ping Phoebe Chen, and Yu-Gang Jiang. Balanced contrastive learning for long-tailed visual recognition. InProceedings of the IEEE/CVF Conference on Computer Vision and Pattern Recognition, pp. 6908–6917, 2022.

---

> ### Author Response · Authors · 2023-11-21
>
> Dear Reviewer yXjC,
>
> We sincerely thank you again for your great efforts in reviewing this paper. We have gone through your points one-by-one and tried to address them carefully. Please don’t hesitate to let us know if you have any further questions.
>
> Best regards,
>
> Authors

---

> > ### Comment · Reviewer_yXjC · 2023-12-05
> >
> > The authors solved my concerns and I will raise the score to accept this paper.

---

> ### Author Response · Authors · 2023-11-23
> **A friendly reminder for discussion**
>
> Dear Reviewer yXjC,
>
> The rebuttal phase ends today and we have not yet received feedback from you. We believe that we have addressed all of your previous concerns. We would really appreciate that if you could check our response and updated paper.
>
> Looking forward to hearing back from you.
>
> Best Regards,
>
> The Authors of Paper 130

---

> > ### Comment · Area_Chair_DZpM · 2023-12-04
> >
> > Thanks much for your active interactions!  We have read your extensive rebuttals and reached out to Reviewer #yXjC for discussions.

---

### Author Response · Authors · 2023-11-19
**General Response**

Dear Reviewers,

We would like to express our gratitude once again for dedicating your valuable time to reviewing our paper and providing us with insightful suggestions. In response to your feedback, we have conducted additional evaluations that involve multiple state-of-the-art DLTR approaches, i.e., BBN, PaCo, and BCL. These experiments were performed on various long-tailed datasets, including CIFAR10-LT, CIFAR100-LT, and ImageNet-LT. The newly obtained results are consistent with the observations presented in the original text. Furthermore, we have also provided the additional empirical observations requested by the reviewers and have thoroughly proofread our paper. The revised sections have been highlighted in blue within the uploaded version.

As the deadline for the author-reviewer discussion period is approaching, we kindly request your feedback on whether our response adequately addresses your concerns.

Thank you,

The Authors

---

### Meta-Review · Area_Chair_DZpM · 2023-12-09

**Metareview:**

This paper tackles long-tailed recognition by examining optimization conflicts between classes and achieving pareto optimality from an  multi-objective optimization (MOO) perspective.   By depolying MOO methods with a specialized dynamic re-balancing strategy temporally for long-tailed classification, which allows shared early layer features to change less during later stages of learning, the proposed method enhances the performance of long-tailed methods.

This paper has received three detailed reviews.  Reviewers like the presentation clarity, a gradient-based approach on long-tailed recognition from MOO, no additional computational cost, and universal performance gains.  Reviewers have reservations about the technical novelty, general utility for SOTA long-tailed methods, and the characterization of gradient conflicts in cosine similarity only ignoring gradient magnitudes. The authors respond actively with additional clarification and experimental results, successfully turning reviewers' final ratings to a unanimous weak acceptance.

**Justification For Why Not Higher Score:**

The AC concurs with the consensus of a valuable insight with a simple gradient-based MOO optimization add-on to long-tailed recognition methods.  While performance gains are demonstrated over across a variety of long-tailed methods, they could be very minor.  Based on these strengths and weaknesses, the AC recommends weak acceptance.

**Justification For Why Not Lower Score:**

The AC concurs with the consensus of a valuable insight with a simple gradient-based MOO optimization add-on to long-tailed recognition methods.

---

### Decision · Program_Chairs · 2024-01-16

Accept (poster)